# Layer as Puzzle Pieces: Compressing Large Language Models through Layer Concatenation

Fei Wang[1,2], Li Shen[3,6], Liang Ding[4], Chao Xue[2], Ye Liu[1], Changxing Ding[1,5,*]
[1]South China University of Technology    [2]JD Explore Academy
[3]Shenzhen Campus of Sun Yat-sen University    [4]University of Sydney    [5]Pazhou Lab
[6]Center for AI Theoretical Foundation and Systems, Shenzhen Loop Area Institute
ft_feiw@mail.scut.edu.cn, chxding@scut.edu.cn

## Abstract

Large Language Models excel at natural language processing tasks, but their massive size leads to high computational and storage demands. Recent works have sought to reduce their model size through layer-wise structured pruning. However, they tend to ignore retaining the capabilities in the pruned part. In this work, we re-examine structured pruning paradigms and uncover several key limitations: 1) notable performance degradation due to direct layer removal, 2) incompetent linear weight layer aggregation, and 3) the lack of effective post-training recovery mechanisms. To address these limitations, we propose CoMe, including a progressive layer pruning framework with a **Co**ncatenation-based **Me**rging technology and a hierarchical distillation post-training process. Specifically, we introduce a channel sensitivity metric that utilizes activation intensity and weight norms for fine-grained channel selection. Subsequently, we employ a concatenation-based layer merging method to fuse the most critical channels across adjacent layers, enabling progressive model size reduction. Finally, we propose a hierarchical distillation protocol that leverages the correspondences between the original and pruned model layers established during pruning, thereby enabling efficient knowledge transfer. Experiments on seven benchmarks show that CoMe achieves state-of-the-art performance; when pruning 30% of LLaMA-2-7b's parameters, the pruned model retains 83% of its original average accuracy.[2]

## 1 Introduction

Large Language Models (LLMs) [1, 3, 41, 37, 9] have become the cornerstone of modern natural language processing, enabling start-of-the-art performance in tasks such as text generation [20, 12, 21, 32], machine translation [45], question answering [35, 13], and a variety of other challenging tasks [29, 30, 38]. Their success is primarily attributed to scaling up model parameters, which enhances their representational capacity [15]. However, the rapid growth of model size comes at a cost: LLMs' computational and storage demands have become a significant obstacle for practical deployment, especially in resource-constrained environments.

Recent works resort to model compression [46, 39] to reduce the resource footprint of LLMs, with mainstream approaches encompassing model pruning [24, 34, 27, 16] and knowledge distillation [31, 10, 5]. Among these, structured layer pruning is desirable for its hardware efficiency, as it removes entire modules and reduces computational complexity [24, 34, 27, 16]. While direct layer pruning reduces model size, it often leads to performance degradation. To mitigate this, several studies have proposed merging adjacent layers through linear aggregation of their weights [27, 42, 22]. These

---

*Corresponding author.
[2]Our code is available at https://github.com/MPI-Lab/CoMe.

approaches aim to better preserve model capacity by combining information from multiple layers, yet important questions remain regarding their effectiveness and the underlying assumptions.

Despite the efficiency of layer pruning, we rethink and identify several critical limitations in existing approaches. Direct layer pruning assumes that specific layers are redundant and can be removed without harming model performance. However, our analysis reveals that different methods yield highly inconsistent rankings of layer importance, and pruning different layers leads to performance degradation on different benchmarks. This suggests that each layer contributes meaningfully to the model, and preserving the mapping functions of pruned layers is essential. In addition, linear weight aggregation methods rely on the assumption of distributional similarity among layer weights, which does not hold for feed-forward networks; as a result, linear aggregation can further exacerbate information loss. Finally, most current pruning methods lack integrated post-training recovery schemes. While some studies combine pruning and knowledge distillation, these stages are typically handled separately, resulting in suboptimal knowledge transfer and limited performance recovery.

Motivated by these insights, we propose CoMe, a structured compression framework for LLMs that integrates progressive layer pruning with a **Co**ncatenation-based **Me**rging technique and a hierarchical distillation protocol. Our approach consists of three key components. First, we introduce a channel-level sensitivity metric that quantifies the importance of each channel according to activation response intensity and weight norms, providing a principled basis for fine-grained pruning. Second, we present a progressive merging method that concatenates important channels from adjacent layers, reconstructing compact fusion layers and minimizing information loss. Iteratively applying this strategy yields a pruned model with significantly reduced complexity. Third, we propose a hierarchical distillation protocol that utilizes layer correspondences established during pruning to guide efficient knowledge transfer via decoupled layer-wise feature alignment. This protocol accelerates post-training and reduces computational overhead. Fig. 1 depicts the the overall procedure.

We evaluate CoMe on seven LLMs, three sparsity levels (10%, 20%, 30%), seven NLP benchmarks, and two datasets, comparing against nine competitive baselines. Experimental results show that CoMe consistently outperforms existing methods: (1) During pruning, it achieves superior performance across all settings; (2) After post-training, it surpasses state-of-the-art by over 2.4% in average accuracy; (3) Compared to linear weight aggregation methods, concatenation-based merging improves average accuracy by more than 2% and reduces perplexity by over 4.7 across benchmarks.

In summary, our main contributions are as follows:

- We introduce CoMe, a novel framework for structured layer pruning and recovery in LLMs, which preserves critical channels and enables efficient post-training restoration.

- We propose a concatenation-based merging method that reconstructs layers using the most informative channels, which minimizes performance loss caused by pruning.

- We develop a hierarchical distillation protocol that exploits pruning priors to efficiently transfer knowledge through decoupled layer-wise feature alignment.

- We conduct extensive experiments, reveal the inherent limitations of linear weight aggregation in layer pruning, and demonstrate the effectiveness of our approach across multiple LLMs and benchmarks.

## 2 Related Work

**Structured Layer Pruning.** LLMs are typically constructed by stacking multiple transformer layers, and substantial research has been devoted to compressing these models via layer pruning. Existing methods can be broadly grouped into three categories. First, *heuristic layer pruning* approaches, such as Magnitude [19, 16] and Taylor [23, 16], evaluate the layer importance based on weight norm or the error induced by removing specific weights. While straightforward, these methods often fail to capture the complex, nonlinear dependencies across layers. Second, *redundancy-based pruning* estimates block importance based on activation patterns. For example, Men et al. [24] identify redundant layers via cosine similarity between output features, and Song et al. [34] select sub-models by evaluating their performance on a calibration set. Although these methods incorporate global performance metrics, their discrete search spaces make it challenging to preserve the essential feature encoding capacities of pruned layers. Third, *layer fusion* methods aim to improve parameter utilization by

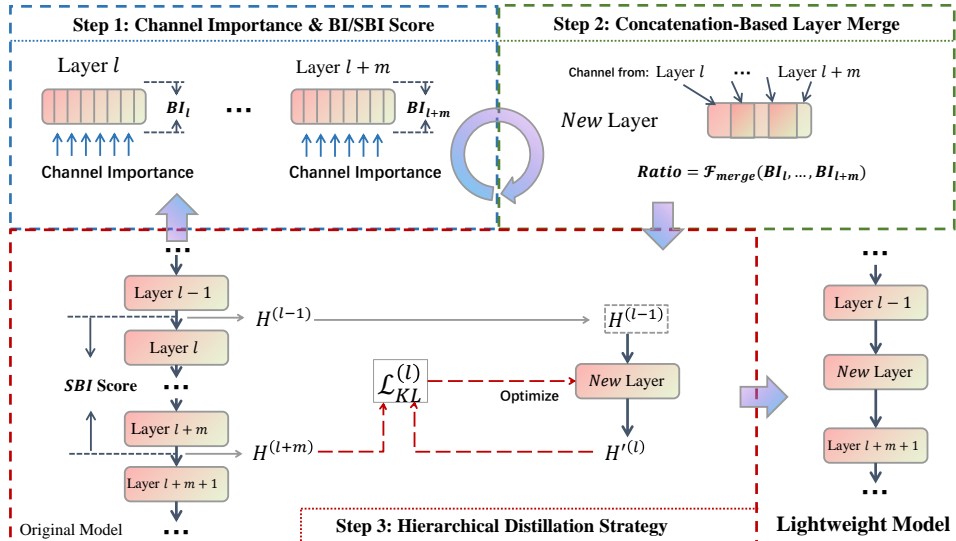

Figure 1: Overview of CoMe. **Steps 1** and **2** are iteratively executed until the model achieves the target size. Following this, **Step 3** utilizes efficient post-training through feature-level distillation to recover the performance. The resulting lightweight model incorporates several newly replaced layers.

reconstructing layers through linear aggregation of weights. Techniques such as MKA [22] use manifold learning to guide layer fusion, while LaCo [42] quantifies parameter differences for better information retention. However, these approaches often overlook the position-sensitive properties of Position-wise Feed-Forward Networks (FFN), leading to suboptimal performance on tasks requiring fine-grained semantic understanding. Despite these advances, current pruning methods are limited by the difficulty of accurately quantifying parameter importance and the lack of effective mechanisms for preserving layer capacity, which restricts further improvements in both compression efficiency and pruned model performance.

**Knowledge Distillation.** Knowledge distillation is widely used to transfer the capabilities of large models to smaller ones, emphasizing efficient and effective knowledge transfer. However, aligning features between layers and ensuring training efficiency remains a significant challenge. Early works such as DistilBERT [33] uses layer-wise sampling to initialize student models, but this approach disrupts inter-layer knowledge flow and requires additional alignment losses during training. More recent methods, including MiniLLM [10] and DistiLLM [17], adopts global feature alignment, simplifying mapping but increasing computational costs due to the involvement of the entire model. LLM-Streamline [5] introduces dynamic layer replacement, yet its reliance on single-layer mappings limits model expressiveness. Overall, current distillation techniques face two main challenges: Insufficient use of structural priors from pruning, which leads to feature mismatching between the pruning and distillation stages, and the lack of hierarchical knowledge transfer results in a trade-off between resource efficiency and performance recovery.

**Our Contributions.** To overcome these challenges, we introduce a channel sensitivity metric for fine-grained assessment of parameter importance, reducing discretization issues in existing pruning strategies. We also propose a progressive channel concatenation strategy as a principled alternative to linear aggregation, which mitigates over-smoothing in layer fusion. For knowledge distillation, our hierarchical protocol leverages layer correspondences established during pruning to facilitate efficient and continuous multi-level feature alignment between the original and pruned models. This framework enhances the efficiency of knowledge transfer in compressed LLMs.

## 3 Rethinking the Layer-based Structured Pruning

Based on how parameters are manipulated, layer pruning methods can be divided into two main paradigms: Direct Layer Pruning (DLP) and Weighted Sum-based Layer Pruning (WSLP). DLP is a coarse-grained approach that removes entire layers based on predefined importance metrics [34, 24,

16]. In contrast, WSLP reduces the number of layers by linearly combining multiple layers into one layer [22, 27, 42]. This section analyzes the limitations of both paradigms from the perspectives of layer importance, pruning performance, and the effectiveness of linear aggregation and motivates the design principles behind CoMe.

***Core Issue 1: Are any layers in LLMs truly "redundant"?*** DLP methods typically rely on importance metrics, such as weight norms or cosine distance, to identify and remove supposedly redundant layers [24, 16]. They implicitly assume that some layers make negligible contributions to overall model performance. To investigate this assumption, we compare layer importance scores from various DLP methods on the LLaMA-2-7b model (Fig. 2). The distributions of importance scores differ markedly across methods, with only a few shallow or deep layers consistently identified as highly important. In some cases, layer importance correlates with depth, while in others, it does not. This

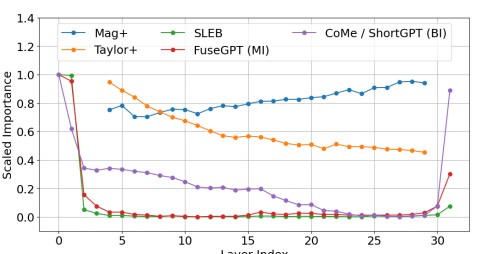

Figure 2: Comparative analysis of normalized layer importance score across different methods. The score of SLEB and FuseGPT is derived from the first-round pruning.

inconsistency indicates that the notion of "redundant" layers depends highly on the metric. Further, DLP methods yield varied performance across benchmarks (please refer to Tab. 8 in the appendix.). For example, pruning 10% of parameters with ShortGPT [24] maintains performance on WinoG and MMLU, but leads to significant degradation on other tasks. No method consistently outperforms others across all benchmarks and sparsity levels. These findings suggest that pruning different layers impairs different capabilities. Removing entire layers risks eliminating intermediate feature mappings that are critical for downstream tasks, which can result in notable performance loss. Therefore, preserving the mapping capacity of pruned layers is crucial for mitigating such degradation.

***Core Issue 2: Does linear weight aggregation preserve hierarchical knowledge and mapping capability?*** WSLP methods merge adjacent layers by linearly aggregating their weights, under the assumption that channel-wise alignment exists not only in Multi-Head Attention (MHA) and normalization modules, but also in Feed-Forward Networks (FFN) [23, 2]. However, while residual connections in Transformers do facilitate channel alignment in MHA and normalization, this property does not extend to FFN modules, which lack explicit feature correspondences across layers [23]. As a result, linear aggregation in FFN modules can lead to oversmoothing weights and disrupt layer-specific knowledge preservation. To empirically examine this issue, we conduct fusion experiments by applying linear aggregation to adjacent layers under various fusion ratios (Figs. 3 and 13). We evaluate the resulting models using perplexity on multiple datasets, considering three module

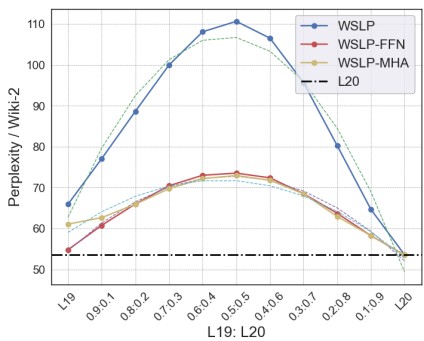

Figure 3: Merge adjacent layers with linear weight aggregation. First, we reduce LLaMA-2-7b's layer number to 23 with ShortGPT. Second, the components in layers 19 and 20 are merged at different ratios. When merging MHA or FFN, the other component is retained from layer 20.

types: FFN, MHA, and the complete Transformer layer. Across all settings, models obtained via linear aggregation consistently underperform those produced by directly pruning layers ("L20"), as indicated by higher perplexity scores. These results confirm that weight distributions in adjacent layers are not sufficiently aligned and that WSLP fails to maintain the original network's hierarchical knowledge and mapping capacity.

In summary, DLP and WSLP have fundamental limitations: DLP risks discarding meaningful intermediate representations, while WSLP introduces over-smoothing and fails to preserve essential knowledge structures. These observations highlight the need for alternative pruning strategies that better retain LLMs' representational and mapping capabilities.

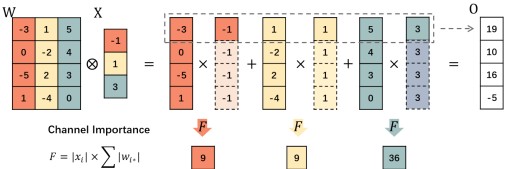

Figure 4: Channel importance calculation process. The importance scores reflect the expected changes of output caused by different channels.

Figure 5: Concatenation-based weight merge process. $W^{(l)}$ and $W^{(l+1)}$ are derived from the same positions in adjacent layers.

## 4 Methodology

This section details the technical innovations of CoMe. During the iterative layer pruning phase, motivated by insights discussed in Section 3, we aim to minimize performance degradation by preserving the most critical parameters from pruned layers. We introduce a Channel Sensitivity Metric that quantifies the influence of individual weight channels on the module's output, thereby facilitating the identification of the most essential channels within layers. Then, we extend the Block Influence (BI) score [24] to the cross-layer setting by introducing the Skip-BI (SBI) score, which enables us to quantify the extent to which a group of adjacent layers alters intermediate features. Based on SBI score, we identify groups of layers that minimally affect feature transformations and implement our concatenation-based merging strategy with the channel importance information. The concatenation-based merging strategy retains the essential channels, thereby mitigating the propagation of disruptive changes from shallow to deeper layers. By iteratively applying the above process, multiple layers are fused into a single layer, thereby progressively pruning the model. In the post-training stage, we further exploit the layer correspondences established during the pruning phase to fine-tune the merged layers in the pruned model. We employ two distinct knowledge distillation strategies, enabling the merged layers to recover the mapping capacities of the original layer groups more effectively. The complete CoMe pipeline is depicted in Fig. 1, and the algorithm framework is shown in Algs. 1, 3 and 4.

### 4.1 Channel Sensitivity Metric

To enable concatenation-based layer merging, we first establish a channel importance metric that quantifies the impact of pruning individual neural pathways. Consider a linear mapping characterized by a weight matrix $W \in \mathcal{R}^{v \times u}$. The input is a vector $X = [x_1, x_2, \ldots, x_u]$, resulting in an output $O = [o_1, o_2, \ldots, o_v]$. We express the linear transformation as $O = W \times X$. As shown in Fig. 4, pruning the weights $W$ at the channel level alters the output $O$. We formulate channel sensitivity through output perturbation analysis. Pruning the $i$-th weight channel (achieved by setting $x_i = 0$) induces perturbation $\Delta O^{(i)} = O - O' = W[:, I] \cdot x_i$. As shown in Fig. 4, we compute the expected $\ell_1$-norm of this perturbation across the calibration dataset $\mathcal{D}$ as the quantizer of channel importance:

$$s_i = \mathbb{E}_{\mathcal{D}}\left[\|\Delta O^{(i)}\|_1\right] = \mathbb{E}_{\mathcal{D}}[|x_i| \sum_{k=1}^{v} |w_{i,k}|], \tag{1}$$

where the expectation over the calibration dataset $\mathcal{D}$ decouples input statistics from static weights. A larger score indicates that the weight channel is more significant, and discarding it would cause more substantial damage to the block's functionality.

### 4.2 Progressive Concatenation-based Layer Merge

In existing work, evaluating the importance of blocks involves analyzing model weights, activations, gradients, and differences in input and output. Men et al. [24] posit that the cosine distance between the hidden features of a block's input and output measures the block's redundancy. Consequently, Men et al. [24] define the Block Influence (BI) score for the $l$-th layer as:

$$BI_l = 1 - \mathbb{E}_{\mathcal{D}} \frac{\mathbf{H}^{(l-1)\mathsf{T}} \mathbf{H}^{(l)}}{\|\mathbf{H}^{(l-1)}\|_2 \|\mathbf{H}^{(l)}\|_2}, \tag{2}$$

where $\mathbf{H}^{(l)} \in \mathbb{R}^{S \times d}$ represents the output hidden features of the $l^{th}$ layer in the model. $S$ is the sequence length, and $d$ is the hidden dimension. $\mathcal{D}$ is the calibration dataset used.

The objective of CoMe is to replace multiple original blocks with fused blocks while minimizing performance loss. Therefore, we extend the BI concept to layer groups as a novel Skip-Block Influence (SBI) metric. For the module containing layers $l$ to $l + m$, its SBI score is defined as:

$$SBI_{l:l+m} = 1 - \mathbb{E}_{\mathcal{D}} \frac{\mathbf{H}^{(l-1)\mathsf{T}}\mathbf{H}^{(l+m)}}{\|\mathbf{H}^{(l-1)}\|_2 \|\mathbf{H}^{(l+m)}\|_2}, \tag{3}$$

where $m + 1 \geq 2$ indicates the number of layers to be merged. A smaller $SBI$ score indicates that the block group spanning multiple blocks induces more minor perturbations to the hidden features, implying that manipulating these groups will have a minor impact on model performance. We factorize block weights into channel-level components, treating channels as atomic units and concatenating the parameters sequentially to increase the importance of each channel. The concatenation-based merge (Fig. 5) implements parameter preservation through:

$$W^{(\text{merge})} = \bigoplus_{k=l}^{l+m} W^{(k)}[:, \mathcal{T}_k], \tag{4}$$

where $\mathcal{T}_k$ denotes the top-k channels selected via our sensitivity metric, and $\oplus$ indicates column-wise concatenation. It ensures the retention of the most impactful parameters in merged layers. For the parameter preservation ratio, we employ a heuristic approach. The parameter preservation ratio for layer $t$ in group $\{l, ..., l + m\}$ follows:

$$r_t = \frac{BI_t^p}{\sum_{i=l}^{l+m} BI_i^p}, \quad p > 0, \tag{5}$$

where $p$ controls the distribution skewness, and a larger $p$ value emphasizes the preservation of layers with high BI scores. Through constrained concatenation where $\sum r_t = 1$, the merge module keeps the hidden state dimension as the size of the origin module. We provide the calculation details for channel importance and the merge rules for different structures in Section C.

## 4.3 Post-Training via Hierarchical Distillation Strategy

After a progressive merging process, we aim to replace the original layers with the fused ones to alleviate representational gaps induced by pruning. Thus, each fused layer should maintain equivalent feature representation capabilities to its original group, enhancing model performance. We align their feature representations through post-training. Our progressive layer merging process creates a direct mapping between pruned and original layers, formalized as: $\mathcal{P} = [\{a_1, b_1\}, \ldots, \{a_N, b_N\}]$, where $a$ is a layer index from origin model, $b$ is from the pruned model and $N$ is the number of merged layers. It provides a priori conditions for efficient feature-based post-training.

We use the original model as the teacher model and the pruned lightweight model as the student model, transferring knowledge from the teacher model to the student model. We employ feature-level distillation using symmetric Kullback-Leibler divergence (KL) to mitigate distribution shift:

$$\mathcal{L}_{\text{KL}} = \mathbb{E}_{\mathcal{D}_{train}}[D_{\text{KL}}\left(\sigma(\mathbf{H}^{(t,a)}) \parallel \sigma(\mathbf{H}^{(s,b)})\right)], \tag{6}$$

where $\mathbf{H}^{(t,a)}$ denotes the output features of the $a$-th layer of the teacher model, and $\mathbf{H}^{(s,b)}$ denotes the output features of the $b$-th layer of the student model. The element pair $\{a, b\} \in \mathcal{P}$. $\sigma$ denotes the softmax activation function. $\mathcal{D}_{train}$ is the training dataset. The KL-divergence term expands as:

$$D_{\text{KL}}(P \parallel Q) = \sum_{i=1}^{|\mathcal{D}_{train}|} P_i \log \frac{P_i}{Q_i}, \quad P_i^{(k)} = \sigma(\mathbf{H}^{(t)}), \quad Q_i^{(k)} = \sigma(\mathbf{H}^{(s)}). \tag{7}$$

The optimization process requires iterating through all mapping pairs in $\mathcal{P}$ from shallow to deep layers. This process is named multi-process post-training (CoMe-mp). While CoMe-mp updates one layer per iteration and uses minimal resources, its sequential nature prevents joint optimization of shallow and deep layers. To address this, we consolidate multiple training processes into one, iterating over

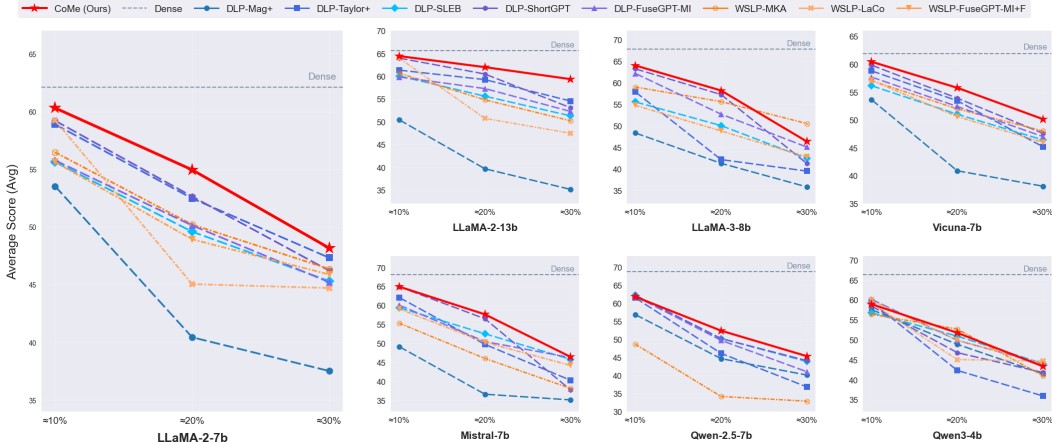

Figure 6: Comparison of different layer-wise pruning, including CoMe (red), DLP (blue series), and WSLP (orange series), on various models with 10%, 20%, and 30% sparsity.

all mapping relationships in $\mathcal{P}$ simultaneously, and name it single-process post-training (CoMe-sp). It minimizes the KL divergence across multiple mappings:

$$\mathcal{L}_{\text{KL-sp}} = \frac{1}{|\mathcal{P}|} \sum_{\{a,b\}\in\mathcal{P}} \mathbb{E}_{\mathcal{D}_{train}}[D_{\text{KL}}\left(\sigma(\mathbf{H}^{(t,a)}) \parallel \sigma(\mathbf{H}^{(s,b)})\right)]. \tag{8}$$

CoMe-sp enables the joint optimization of multiple fusion layers using global information, which helps capture inter-layer dependencies and allows for more coordinated parameter updates. Although CoMe-sp requires more storage for optimization parameters, increasing by a factor of more than $|\mathcal{P}|$ compared to CoMe-mp, it often achieves more effective training and better overall model performance due to its holistic optimization strategy.

## 5 Experiments

### 5.1 Experimental Setting

**Models, Datasets, and Metrics.** We evaluate our method on several widely used open-source models, including LLaMA-2-7b, LLaMA-2-13b [37], and LLaMA-3-8b [9] from the LLaMA series, Qwen-2.5-7b [41], Qwen-3-4b, Mistral-7b [14], and Vicuna-7b [44]. Model performance is assessed using the lm-evaluation-harness [8] framework on seven standard benchmarks commonly adopted in model compression research: ARC-challenge (ARC-c), ARC-easy (ARC-e) [6], HellaSwag (HellaS) [43], OpenBookQA (OBQA) [26], PIQA [4], Winoground (WinoG) [36] under the zero-shot setting, and MMLU [11] under the five-shot setting. Accuracy is reported with normalized option lengths to ensure comparability. We report the average accuracy (Avg) across all datasets and the Retained Performance (RP), defined as the percentage of the original model's accuracy preserved after pruning. Perplexity (PPL) is measured on the C4 [28] and Wikitext-2 (Wiki-2) [25] datasets.

**Baselines.** Eight state-of-the-art layer pruning methods are selected as baselines. DLP-based methods include Magnitude+ (Mag+) [19, 16], Taylor+ [23, 16], ShortGPT [24], SLEB [34], and FuseGPT-MI [27]. WSLP-based methods include FuseGPT-MI-F [27], MKA [22], and LaCo [42]. For a fair comparison, unless otherwise noted, the post-training phase is excluded from FuseGPT. FuseGPT-MI refers to pruning with Macro Influence (MI) without fusion, while FuseGPT-MI-F includes layer fusion. Tables distinguish between DLP and WSLP methods with horizontal lines. For post-training comparisons, LLM-Streamline [5] and FuseGPT with training are also evaluated. Detailed descriptions of all baseline methods are provided in Section A.

**Implementation.** All methods are evaluated under sparsity levels of 10%, 20%, and 30%. Tab. 3 summarizes model configurations for each sparsity setting, while Tab. 4 lists the pruned layer sequences for all DLP methods. Unless otherwise specified, CoMe is conducted on LLaMA-2-7b using the Wiki-2 calibration set (256 samples) with a default sparsity of 30%. During the layer

Table 1: The Post-training Experiment on the LLaMA-2-7b and Qwen3-4b.

| Method | | ARC-c | ARC-e | HellaS | Benchmark↑
OBQA | PIQA | WinoG | MMLU (5) | Avg↑ | RP↑ | PPL↓
C4 | Wiki-2 |
|---|---|---|---|---|---|---|---|---|---|---|---|---|
| LLaMA-2-7b Dense | | 46.33 | 74.54 | 75.99 | 44.20 | 79.05 | 69.06 | 45.60 | 62.11 | 100.00 | 7.27 | 5.47 |
| Prune (30.0%) | FuseGPT-MI | 30.20 | 50.59 | 52.98 | 33.60 | **69.37** | 54.54 | 25.17 | 45.21 | 71.53 | **17.60** | 14.94 |
| | FuseGPT-MI+F | 30.20 | 50.13 | 55.08 | 34.80 | 68.50 | 55.41 | 27.01 | 45.88 | 72.82 | 17.80 | **14.34** |
| | LLM-streamline | 33.79 | 45.33 | 50.94 | 31.60 | 63.49 | **63.14** | **41.82** | 47.16 | **76.53** | 70.10 | 65.84 |
| | CoMe | **35.24** | **54.46** | **56.56** | **35.40** | 68.88 | 61.17 | 25.50 | **48.17** | 76.47 | 19.93 | 16.53 |
| w/ Post-training | FuseGPT-MI | 30.55 | 56.52 | 55.42 | 36.6 | 71.11 | 54.46 | 24.95 | 47.09(↑1.88) | 74.43 | 13.31 | 9.74 |
| | FuseGPT-MI+F | 31.23 | 57.41 | 56.26 | 35.6 | 71.22 | 55.8 | 26.72 | 47.75(↑1.87) | 75.50 | 12.96 | 8.85 |
| | LM-streamline | **36.35** | 51.43 | 56.40 | 33.60 | 66.00 | **66.61** | 40.50 | 50.13(↑2.97) | 80.92 | 18.60 | 19.75 |
| | CoMe-mp | 35.24 | 60.65 | 61.14 | 37.80 | 70.57 | 64.56 | 25.35 | 50.76(↑2.61) | 80.25 | 13.01 | 9.72 |
| | CoMe-sp | 35.58 | **63.51** | **65.83** | **39.20** | **74.05** | 63.38 | 26.48 | **52.58**(↑4.41) | **82.98** | **11.45** | **8.54** |
| Qwen3-4b Dense | | 51.54 | 76.43 | 73.70 | 41.20 | 77.80 | 71.03 | 73.01 | 66.39 | 100.00 | 13.31 | 7.90 |
| Prune (30.1%) | LLM-streamline | 26.71 | 40.57 | 38.31 | 28.20 | 61.81 | **53.91** | 23.54 | 39.01 | 58.99 | 200.93 | 228.89 |
| | CoMe | **28.67** | **47.10** | **43.95** | **29.60** | **63.00** | 51.46 | **32.67** | 42.35 | **63.84** | **56.13** | **37.14** |
| w/ post training | LLM-streamline | 27.22 | 45.37 | 44.20 | 30.40 | **66.59** | 56.99 | 22.96 | 41.96(↑2.95) | 63.32 | 34.80 | 35.77 |
| | CoMe-mp | 30.97 | 51.98 | 47.02 | 30.80 | 65.23 | 54.54 | 36.44 | 45.28(↑2.93) | 68.17 | 32.02 | 22.09 |
| | CoMe-sp | **31.40** | **53.91** | **48.33** | **33.80** | 65.23 | 56.75 | **41.10** | **47.22**(↑4.87) | **71.30** | **29.28** | **20.25** |

pruning process, we set the number of layers merged per iteration to 2 (i.e., $m = 1$). Comprehensive experimental settings and implementation details are available in Section B.

## 5.2 Main Result

**CoMe Outperforms DLP and WSLP Across Benchmarks.** Fig. 6 summarizes the average accuracy of various layer pruning methods. In most settings, CoMe consistently ranks first. When pruning 30% of the parameters from LLaMA-3-8b, CoMe achieves the second-highest average accuracy. When pruning Qwen-3-4b, CoMe achieves a comparable performance compared to the best method. These results indicate that CoMe effectively preserves model capability and enhances the performance of pruned models. Notably, the performance of CoMe declines approximately linearly as sparsity increases, suggesting that important parameters are selectively retained even at high sparsity levels. On the larger LLaMA-2-13b model with 30% sparsity, CoMe surpasses all other methods by at least 4% in average accuracy, highlighting its particular advantage for large-scale models.

**CoMe-sp Surpasses Existing Post-Training Layer-Based Methods.** Tab. 1 presents the effectiveness of various post-training strategies in restoring performance after layer pruning. CoMe achieves the highest average accuracy after pruning, outperforming LLM-Streamline by 1% on LLaMA-2-7b and by 2.3% on Qwen-3-4b, and exceeding FuseGPT-MI+F by 2%. Its PPL is also comparable to the best-performing method. After post-training, the relative ranking of methods remains consistent, with CoMe-sp achieving the top performance, surpassing LLM-Streamline by 2.4% on LLaMA-2-7b, and by 5.2% on Qwen-3-4b, while also achieving lower PPL. These results indicate that the hierarchical distillation strategy in CoMe-sp accelerates performance recovery, even with limited training data (Tab. 7). In contrast, CoMe-mp, which relies only on local layer feature alignment, performs similarly to LLM-Streamline, indicating that relying solely on local layer feature alignment is insufficient for restoring model performance effectively.

## 5.3 Ablation Study

**Ablation on Concatenated Structure.** The results shown in Figs. 11 and 14 indicate that using the CoMe to merge the MHA module, FFN module, or entire layer structure of adjacent layers can achieve better PPL metrics at specific fusion ratios compared to simply retaining layer 20. It demonstrates that the concatenation-based layer merging strategy effectively preserves the model's language modeling capability. Notably, merging the entire layer structure outperforms merging the MHA or FFN modules individually. It indicates that the collaborative fusion of MHA and FFN modules can produce a complementary effect on the parameters. The results validate the effectiveness of the fusion ratio allocation strategy.

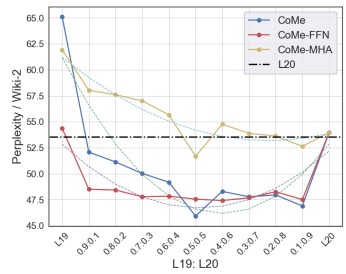

Figure 11: Merge layers with CoMe. The setting is the same as Fig. 3.

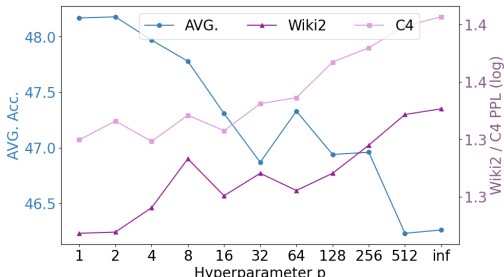

Figure 7: Effect of $p$ in heuristic merge ratio. As $p$ grows, more parameters from layers with higher BI scores are merged.

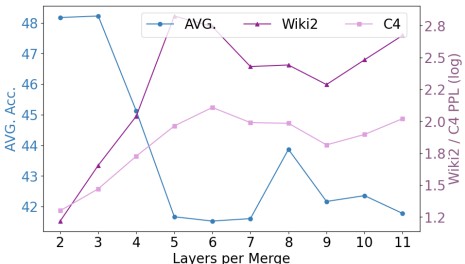

Figure 8: Impact of merge step granularity. Merging more layers degrades performance due to parameter distribution differences.

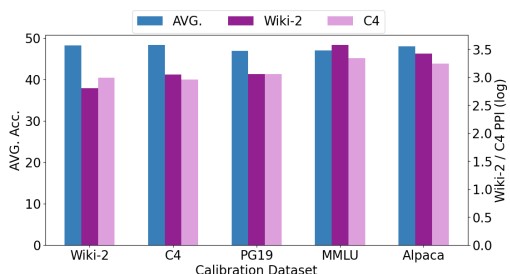

Figure 9: Impact of the calibration dataset. We use two samples in PG19 and 256 in the others.

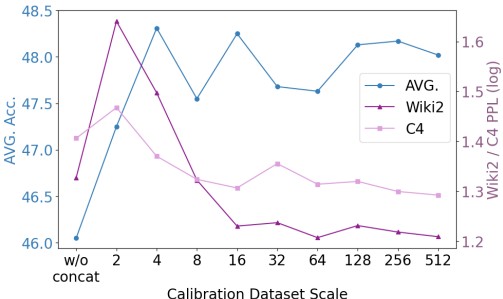

Figure 10: Effect of calibration data scale. CoMe achieves optimal performance with samples more than 128.

**Effectiveness of Concatenation-Based Merge.** As shown in Figs. 11 and 14, merging adjacent layers using the concatenation-based strategy in CoMe yields lower PPL than simply retaining layer 20, demonstrating superior language modeling capability. Merging the entire layer structure consistently outperforms merging only the MHA or FFN modules, suggesting that the joint fusion of MHA and FFN produces a complementary effect. However, the optimal fusion ratio varies across the three types of structures, suggesting that the heuristic fusion ratio allocation strategy has limitations and may not fully unlock the potential of CoMe.

**Impact of $p$ in Heuristic Merge Ratio.** Fig. 7 illustrates the impact of $p$ on the performance of models pruned using CoMe. For $p \leq 2$, the model maintains optimal performance after pruning, with average accuracy exceeding 48%. As $p$ increases, retention becomes more skewed toward layers with higher BI scores, leading to performance degradation. When $p \rightarrow \inf$ (i.e., without merging), performance drops sharply (PPL rises to 25.50/ 21.21). These findings confirm that: (1) the channel sensitivity-based selection mechanism effectively identifies key parameters; (2) progressive layer merging is essential for maintaining performance; and (3) BI scores are positively correlated with the number of essential parameters, validating the skewness control strategy.

**Effect of Merge Step Granularity.** The analysis in Fig. 8 reveals that merging more than two layers at a time leads to a drop in average accuracy of over 3% and a substantial increase in PPL. This suggests that merging multiple disparate layers increases parameter distribution discrepancies. The progressive merging of fewer layers helps mitigate this issue and preserves model performance better.

**Robustness to Calibration Dataset.** Cross-dataset experiments in Fig. 9 demonstrate that CoMe is robust to the choice of calibration dataset. When using pre-training style data, such as Wiki-2, C4, the model achieves low PPL and strong language modeling performance. In contrast, using QA datasets such as MMLU, which differ significantly from the pre-training dataset, increases PPL by over 25%. However, the impact on downstream task average accuracy remains limited (fluctuation < 1.5%), indicating that the channel importance measurement mechanism effectively decouples input distribution from static weight features. These suggest that calibration data similar to the training distribution is preferable.

Table 2: Comparison of CoMe and WSLP, *w/* and *w/o* the merge process. For the methods *w/o Add*, we retain the layers deemed most important by the method (LaCo *w/o Add* retains shallow layers).

| Method | ARC-c | ARC-e | HellaS | Benchmark↑ (↓) OBQA | PIQA | WinoG | MMLU (5) | Avg↑(↓) | RP↑(↓) | PPL↓(↑) C4 | Wiki-2 |
|---|---|---|---|---|---|---|---|---|---|---|---|
| MKA | 34.04 | 49.58 | 48.12 | 35.00 | 63.00 | 59.12 | 35.64 | 46.36 | 75.14 | 810.04 | 455.34 |
| w/o Add | 33.96(-0.08) | 49.45(-0.13) | 48.02(-0.10) | 34.80(-0.20) | 62.79(-0.21) | 59.04(-0.08) | 35.64(-) | 46.24(-0.11) | 74.95(-0.19) | 809.74(-0.30) | 454.70(-0.64) |
| LaCo | 30.97 | 49.79 | 50.14 | 35.00 | 68.34 | 53.91 | 24.84 | 44.71 | 71.11 | 39.18 | 42.67 |
| w/o Add | 30.80(-0.17) | 50.38(+0.59) | 50.90(+0.76) | 34.60(-0.40) | 69.21(+0.87) | 55.01(+1.10) | 24.71(-0.13) | 45.09(+0.37) | 71.53(+0.42) | 25.51(-13.67) | 21.18(-21.49) |
| FuseGPT | 30.20 | 50.13 | 55.08 | 34.80 | 68.50 | 55.41 | 27.01 | 45.88 | 72.82 | 17.80 | 14.34 |
| w/o Add | 30.20(-) | 50.59(+0.46) | 52.98(-2.10) | 33.60(-1.20) | 69.37(+0.87) | 54.54(-0.87) | 25.17(-1.84) | 45.21(-0.67) | 71.53(-1.29) | 17.60(-0.20) | 14.94(+0.59) |
| CoMe | 35.24 | 54.46 | 56.56 | 35.40 | 68.88 | 61.17 | 25.50 | 48.17 | 76.47 | 19.93 | 16.53 |
| w/o Concat | 30.97(-4.27) | 48.40(-6.06) | 55.81(-0.75) | 33.00(-2.40) | 68.12(-0.76) | 59.35(-1.82) | 26.71(+1.21) | 46.05(-2.12) | 72.94(-3.53) | 25.48(+5.54) | 21.26(+4.73) |

**Calibration Data Scale.** As shown in Fig. 10, channel importance stabilizes when calibration samples exceed 128 (PPL fluctuation on WikiText-2 is less than 1.3). When fewer than eight samples are used, parameter merging degenerates into random selection, with PPL exceeding that of the no-fusion strategy ($p = \inf$ in Fig. 7). This highlights the necessity of accurate channel importance evaluation, as incorrect estimates can impair pruning performance.

## 5.4 Weight Sum-Based Merge vs. Concatenation-Based Merge

Both WSLP and CoMe aim to mitigate the loss of layer mapping functionality caused by DLP. Tab. 2 compares these approaches without the merging process. For WSLP, removing the merge step changes average accuracy by less than 0.7% and RP by less than 1.3%, indicating a negligible effect. In contrast, removing the concatenation-based merge in CoMe leads to a significant drop in average accuracy and RP (both decrease by more than 2%) and an increase in PPL (by more than 4.7), demonstrating that the concatenation-based merge plays a critical role in preserving model performance during pruning. Further analysis is provided in Section F.

## 6 Conclusion

This paper addresses the challenge of layer pruning in LLMs, focusing on preserving model performance while reducing computational complexity. Our proposed framework, CoMe, introduces three key innovations: a channel sensitivity metric to quantify the importance according to activation and weight, a concatenation-based merging strategy to retain critical information during pruning effectively, and a hierarchical distillation protocol for efficient post-training recovery. Extensive experiments across multiple models, sparsity levels, and benchmarks demonstrate that CoMe achieves superior performance compared to existing pruning approaches in maintaining accuracy after compression.

## Limitations

CoMe adopts a heuristic and uniform parameter preservation ratio for merging all Transformer components, which limits its adaptability to different architectures. In Section D, we present a posterior-based solution for adaptively merging two adjacent layers to alleviate the limitations of CoMe. In future work, we will explore adaptive methods for merging multiple layers at once and extend CoMe to expert merge.

## Acknowledgements

This work was supported by the Guangdong Provincial Key Field R&D Program Project (2024B0101040004), the National Natural Science Foundation of China under Grants 62476099, 62076101, and 62576364, the Guangdong Basic and Applied Basic Research Foundation under Grants 2024B1515020082 and 2023A1515010007, the Guangdong Provincial Key Laboratory of Human Digital Twin under Grant 2022B1212010004, and the TCL Young Scholars Program.

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

Table 3: Model Parameters and Sparsity Settings Across Different Models.

| Model | Size | # Blocks | ≈10% Sparisity | # Removed | # Parameters | ≈20% Sparisity | # Removed | # Parameters | ≈30% Sparisity | # Removed | # Parameters |
|---|---|---|---|---|---|---|---|---|---|---|---|
| LLaMA-2-7B | 6.7b | 32 | 9.01% | 3 | 6.1b | 21.02% | 7 | 5.3b | 30.03% | 10 | 4.7b |
| LLaMA-2-13B | 13b | 40 | 9.75% | 4 | 11.7b | 19.49% | 8 | 10.5b | 29.24% | 12 | 9.2b |
| LLaMA-3-8b | 8b | 32 | 10.86% | 4 | 7.2b | 19.01% | 7 | 6.5b | 29.88% | 11 | 5.6b |
| Vicuna-7b | 6.7b | 32 | 9.18% | 3 | 6.1b | 21.02% | 7 | 5.3b | 30.03% | 10 | 4.7b |
| Mistral-7B | 7.2b | 32 | 9.04% | 3 | 6.6b | 21.08% | 7 | 5.7b | 30.12% | 10 | 5.1b |
| Qwen-2.5-7b | 7.6b | 28 | 9.18% | 3 | 6.9b | 21.42% | 7 | 6.0b | 30.60% | 10 | 5.3b |
| Qwen-3-4b | 4.0b | 36 | 10.04% | 4 | 3.6b | 20.07% | 8 | 3.22b | 30.11% | 12 | 2.8b |

# A  The Details of Comparison Methods

This paper compares three types of structured pruning paradigms: (1) DLP, which includes Mag+ [16], Taylor+ [16], ShortGPT [24], and SLEB [34]; (2) WSLP, which includes LaCo [42] and MKA [22]; and (3) methods combining layer pruning with post-training, such as LLM-streamline [5] and FuseGPT [27]. We implemented each method using publicly available code.

**Magnitude+ (Mag+).** [3]  Kim et al. [16] use this method as a baseline in the pruning method comparison conducted. Initially proposed by Li et al. [19], it assumes that weights with smaller norms contain less information. For block-level analysis, the importance of the $l$-th layer is calculated as the sum of the first-order norms of all weight parameters: $I_{Mag+,l} = \sum_{W \in \mathcal{W}^{(l)}} \sum_{w \in W} |w|$, where $\mathcal{W}^{(l)}$ represents the set of all weight matrices in the $l$-th layer. We follow popular heuristic algorithms [7, 18]. Kim et al. [16] further mitigated performance degradation by retaining the first four and last two layers [23].

**Taylor+.** [3] This method is also a baseline in the pruning method comparison by Kim et al. [16]. It assumes that the error introduced by removing weight parameters indicates their importance. Given a calibration dataset $\mathcal{D}$, this error can be expressed as a change in the training loss $\mathcal{L}$: $|\mathcal{L}(W; \mathcal{D}) - \mathcal{L}(W = 0; \mathcal{D})| \approx \frac{\partial \mathcal{L}(\mathcal{D})}{\partial W} W$. Following Kim et al. [16] and Ma et al. [23], we define the layer importance parameter as $I_{Taylor+,l} = \sum_{W \in \mathcal{W}^{(l)}} \sum_{w \in W} |\frac{\partial \mathcal{L}(\mathcal{D})}{\partial w} w|$. We use similar heuristic optimization methods to retain the first four and last two layers.

**ShortGPT.** [4] Proposed by Men et al. [24], this method assumes redundancy in the model layers and defines redundancy as layers that minimally alter the hidden embeddings. To measure change, they use the cosine distance as a metric, as shown in Eq. (2). Men et al. [24] use the PG19 long-document dataset for calibration, and we control the size of the calibration dataset to 256 samples. The results are consistent with those reported by Men et al. [24]. Men et al. [24] compute the importance scores for models in the LLaMA family, and we directly use the layer importance order provided in the paper. When the number of pruned layers exceeds the required number, we append the least important layers from our calculated importance scores. For models where Men et al. [24] do not provide layer importance sorting, we estimate it using the settings in this paper.

**SLEB.** [5] Song et al. [34] propose using a posterior method to verify the redundancy of specific layers. SLEB uses the exponential part of the PPL score of the pruned model on a specified dataset as the redundancy score: $I_{SLEB,l} = \sum_{X \in \mathcal{D}} -\frac{1}{K} \sum_{i=0}^{n} \log p_{M'_l}(x_i | x_{<i}, x_i \in X)$, where $M'$ denotes the smaller model obtained from previous pruning steps, $M'_l$ denotes the model obtained after pruning the $l$-th layer, and $X = x_1, ..., x_i, ..., x_n$ represents a sample. $I_{SLEB,l}$ is the exponential part of the PPL score, which positively correlates with the PPL score; hence, $I_{SLEB}$ positively correlates with the PPL score. The SLEB method is a progressive structural search optimized for the PPL on the specified dataset.

**LaCo.** [6]  Proposed by Yang et al. [42], this method uses the weight differences between layers as important information for layer retention. LaCo groups several adjacent layers and performs a Reserving-Differences-while-Seeking-Common layer merge. For weight fusion from layer $l$ to $l + m$, the fused weight is represented as $W^* = W^{(l)} + (W^{(l+1)} - W^{(l)}) + ... + (W^{(l+m)} - W^{(l)}) = W^{(l)} + \sum_i^m (W^{(l+i)} - W^{(l)})$. It fuses the differences between deeper and shallow layers into the shallow layers. LaCo assesses the redundancy of pruned groups using the cosine similarity of output

---

[3] https://github.com/Nota-NetsPresso/shortened-llm
[4] https://github.com/sramshetty/ShortGPT
[5] https://github.com/jiwonsong-dev/SLEB
[6] https://github.com/yangyifei729/LaCo

Table 4: The layer importance ranking of different DLP methods.

| Model | LLaMA-2-7B | | | LLaMA-2-13B | | | LLaMA-3-8b | | |
| --- | --- | --- | --- | --- | --- | --- | --- | --- | --- |
| Sparsity | 9.0% / 21.0% / 30.0% | | | 9.8% / 19.5% / 29.2% | | | 10.9% / 19.0% / 29.9% | | |
| Mag+ | 7, 6, 11 / 8, 4, 10, 9 / 12, 14, 13 | | | 4, 5, 6, 7 / 10, 8, 9, 13 / 12, 11, 14, 16 | | | 5, 8, 7, 11 / 4, 6, 10 / 9, 13, 12, 14 | | |
| Taylor+ | 29, 28, 27 / 26, 21, 25, 23 / 24, 19, 20 | | | 37, 35, 34, 36 / 33, 28, 26, 29 / 32, 27, 31, 25 | | | 29, 28, 26, 25 / 19, 27, 23 / 24, 20, 18, 22 | | |
| ShortGPT | 27, 26, 25 / 28, 24, 29, 23 / 21, 22, 30 | | | 33, 31, 32, 30 / 29, 34, 28, 35 / 27, 26, 36, 37 | | | 25, 27, 26, 24 / 28, 23, 22 / 29, 21, 20, 19 | | |
| SLEB | 14, 23, 11 / 24, 10, 27, 15 / 21, 25, 8 | | | 33, 29, 12, 13 / 26, 31, 14, 32 / 11, 10, 25, 35 | | | 10, 26, 11, 12 / 9, 23, 19 / 22, 25, 8, 7 | | |
| FuseGPT-MI | 11, 8, 27 / 24, 22, 14, 21 / 10, 13, 23 | | | 33, 29, 12, 10 / 27, 35, 31, 30 / 15, 28, 16, 25 | | | 10, 26, 25, 11 / 9, 8, 19 / 22, 7, 23, 20 | | |

| Model | Vicuna-7b | | | Mistral-7B | | | Qwen-2.5-7b | | | Qwen-3-4b | | |
| --- | --- | --- | --- | --- | --- | --- | --- | --- | --- | --- | --- | --- |
| Sparsity | 9.0% / 21.0% / 30.0% | | | 9.0% / 21.1% / 30.1% | | | 9.2% / 21.4% / 30.6% | | | 10.0% / 20.1% / 30.1% | | |
| Mag+ | 7, 6, 11 / 8, 9, 10, 4 / 12, 14, 13 | | | 4, 6, 5 / 12, 7, 9, 10 / 11, 8, 13 | | | 9, 14, 17 / 16, 15, 13, 7 / 12, 6, 10 | | | 21, 19, 20, 18 / 22, 17, 15, 16 / 14, 23, 9, 13 | | |
| Taylor+ | 29, 26, 21 / 27, 24, 25, 23 / 22, 19, 20 | | | 16, 28, 15 / 17, 29, 14, 13 / 22, 18, 12 | | | 4, 5, 21 / 22, 20, 23, 18 / 19, 17, 16 | | | 26, 25, 27, 29 / 28, 24, 23, 22 / 21, 30, 20, 31 | | |
| ShortGPT | 27, 25, 28 / 29, 24, 26, 23 / 22, 21, 30 | | | 25, 26, 24 / 27, 22, 23, 28 / 21, 29, 30 | | | 16, 17, 15 / 14, 12, 13, 18 / 11, 25, 24 | | | 29, 26, 27, 31 / 32, 33, 28, 25 / 20, 16, 18, 30 | | |
| SLEB | 10, 27, 14 / 23, 11, 12, 24 / 13, 9, 26 | | | 14, 13, 15 / 27, 22, 8, 24 / 23, 11, 21 | | | 16, 15, 17 / 14, 13, 18, 12 / 11, 10, 9 | | | 16, 15, 14, 17 / 18, 2, 19, 32 / 21, 26, 11, 30 | | |
| FuseGPT-MI | 12, 27, 11 / 23, 10, 25, 24 / 21, 9, 8 | | | 13, 10, 14 / 11, 8, 27, 23 / 22, 26, 25 | | | 16, 19, 17 / 18, 21, 14, 15 / 22, 10, 13 | | | 16, 17, 15, 2 / 14, 20, 21, 18 / 10, 26, 32, 11 | | |

features between the pruned and unpruned models: $I_{LaCo} = \frac{1}{N} \sum_{X \in \mathcal{D}} \frac{H_M^{(L)\top} H_{M'}^{(L')}}{\|H_M^\top\|_2 \|H_{M'}^\top\|_2}$, where $H_M^{(L)}$

and $H_{M'}^{(L')}$ represent the output features of the last layer of the model. Due to the threshold adjustment for cosine similarity in LaCo and the need to adjust the starting and ending layers for pruning, as well as the number of layers in each group, the excessive parameter settings made it challenging to optimize performance for each model. Therefore, we implement this method only on models in the LLaMA-2 family.

**MKA.** [7] Proposed by [22], this method uses manifold learning and the Normalized Pairwise Information Bottleneck (NPIB) measures to assess layer similarity and fusion. MKA progressively fuses deeper into shallower layers, merging the last two adjacent layers each time. In the code implementation, we find that MKA calculates the NPIB scores for two layers as approximately equal: $I_{NPIB,l} : I_{NPIB,l+1} \approx 0.5 : 0.5$. An exponential mapping increases the fusion proportion of the shallower $l$-th layer: $I_{MKA,l} = \frac{e^{I_{norm}}}{1 - e^{I_{norm}}}$, where $I_{norm} = \frac{I_{NPIB,l}}{I_{NPIB,l} + I_{NPIB,l+1}}$ is the normalized NPIB score: $I_{MKA,l+1} = 1 - I_{MKA,l}$. After mapping, the similarity ratio between the two layers approaches $I_{MKA,l} : I_{MKA,l+1} \approx 0.6 : 0.4$.

**FuseGPT.** [8] Proposed by Pei et al. [27], FuseGPT hypothesizes that layer pruning causes performance loss and uses FFN parameter fusion to integrate layer capabilities into adjacent blocks, as Pei et al. [27] hypothesizes that FFN layers concentrate the main capabilities. Low-rank learnable weight matrices disperse the capabilities of pruned layers, optimizing multiple layers at once to reduce the gap caused by pruning. To better study the effectiveness of fusion, we remove the parameter adjustment part of FuseGPT in pure pruning experiments, using randomly initialized low-rank matrix products to fuse weights. In post-training comparison experiments, we use the complete FuseGPT method. Pei et al. [27] propose a Macro Influence (MI) score to measure the global-level impact of removing a model layer: $I_{MI} = 1 - \frac{1}{N} \sum_{X \in \mathcal{D}} \frac{H_M^{(L)\top} H_{M'}^{(L')}}{\|H_M^\top\|_2 \|H_{M'}^\top\|_2} = 1 - I_{LaCo}$.

**LLM-streamline.** [9] Proposed by Chen et al. [5], LLM-streamline uses $SBI$ (Eq. (3)) to measure redundancy of multiple consecutive layers, replacing these layers with the shallowest layer among them, and fine-tuning this shallowest layer post-training to restore model performance.

# B  The Details of Experiment Setting

The settings for the experiment methods follow mainly those in the original papers. All experiments are conducted using an A100-40G GPU. We conducted pruning experiments on LLaMA-2-7b [10],

---

[7] https://github.com/SempraETY/Pruning-via-Merging
[8] https://github.com/jarvispei/fusegpt
[9] https://github.com/RUCKBReasoning/LLM-Streamline
[10] https://huggingface.co/meta-llama/Llama-2-7b-hf

Table 5: Experimental setting for pruning methods. † idenote methods whose hyperparameters were adjusted to satisfy the sparsity ratio constraints in our implementation. Complete implementation details are documented in Subsection B.1.

| Methods | Calibration | # data | seed |
|---|---|---|---|
| Mag+ | Wiki2 | 128 | 10 |
| Taylor+ | Wiki2 | 128 | 10 |
| ShortGPT† | PG19 | 256 | 10 |
| SLEB | Wiki2 | 128 | 10 |
| FuseGPT | Wiki2 | 32 | 10 |
| MKA† | MMLU | 50 subtask * 5 | 10 |
| LaCo† | Mouron () is a commune in the Arde
Torreorgaz is a municipality in the
The 81st Mechanised Brigade () is a mechanised brigade of the Romanian Land Force
There are 18 National Natural Landmarks in the U.S. state of Washington, out of nearly
Copa Libertadores 1973 was won by defending champions Independiente of A | | |
| CoMe | Wiki2 | 256 | 10 |

Table 6: The Hyper-parameter used in LaCo [42]. $\mathcal{C}$ is the number of layers to be merged during each merging optimization. $\mathcal{I}$ is the minimum interval of layers between two merging operations. $\mathcal{L}$ and $\mathcal{H}$ are the minimum and maximum indices of the range of layers for merging. $\mathcal{T}$ is a similarity threshold.

| | Sparisity | $\mathcal{C}$ | $\mathcal{L}$ | $\mathcal{H}$ | $\mathcal{I}$ | $\mathcal{T}$ |
|---|---|---|---|---|---|---|
| LLaMA-2-7b | 9.01% | 4 | 1 | 32 | 2 | 0.85 |
| | 21.02% | 8 | 1 | 32 | 2 | 0.65 |
| | 30.03% | 6 | 1 | 32 | 2 | 0.55 |
| LLaMA-2-13b | 9.75% | 5 | 1 | 40 | 2 | 0.85 |
| | 19.49% | 5 | 1 | 40 | 2 | 0.70 |
| | 29.24% | 5 | 1 | 40 | 2 | 0.55 |

LLaMA-2-13b [11], LLaMA-3-8b [12], Vicuna-7b [13], Mistral-7b [14], Qwen-2.5-7b [15], and Qwen-3-4b [16] and performed post-training experiments on LLaMA-2-7b and Qwen-3-4b.

We modify some settings based on the original implementations and develop an open-source project with multiple pruning methods. Our project code can be found at https://github.com/MPI-Lab/CoMe.

## B.1 Implementation of Pruning Methods

Tab. 5 shows the calibration datasets, the dataset number, and the random seeds used in the pruning methods. Tab. 4 presents the pruned layers' index order for the pruning method. We implement the Mag+, Taylor+, and SLEB using our reproduced code.

For the ShortGPT method, we follow the layer BI score for LLaMA-2-7B provided in the original article. For the LLaMA-2-13b model, the original paper provides only the pruning order for the first 10 layers. We use the open-source project reproduction code to calculate the remaining layers' BI scores and place the two with the smallest BI scores at the end of the given pruning order. For other models, we obtain the BI scores for each layer entirely through the reproduction method. PG19 is a long-document dataset, and the training set contains 28,602 training samples. Using all samples to get the model's BI scores would consume significant training resources, so we randomly selected 256 training samples from PG19 for calibration. Even with a small amount of data, the ShortGPT method takes much longer to calculate BI scores than other methods.

---

[11] https://huggingface.co/meta-llama/Llama-2-13b-hf

[12] https://huggingface.co/meta-llama/Meta-Llama-3-8B

[13] https://huggingface.co/lmsys/vicuna-7b-v1.5

[14] https://huggingface.co/mistralai/Mistral-7B-v0.1

[15] https://huggingface.co/Qwen/Qwen2.5-7B

[16] https://huggingface.co/Qwen/Qwen3-4B-Base

When selecting MMLU data, MKA randomly samples five samples from 50 sub-tasks. In our implementation, we uniformly sample 250 samples from each sub-task. We reproduce the experimental results for the LLaMA family using the original MKA code, while we obtain the sparse results for other models using our reproduced code. The Qwen-2.5-7b model contains bias weights, and fusing these weights would degrade the performance of the pruned model, so we do not fuse the bias weights.

The calibration samples for the LaCo method are sourced from the open-source project code, and we fully reproduce the process using the original code. To achieve the number of pruned layers consistent with the settings in this paper, we make simple parameter adjustments to the LaCo method, with the detailed parameter settings shown in Tab. 6. For other models, adjusting the LaCo code is too complex, so we do not reproduce it.

The FuseGPT method is implemented using the original code. To compare different categories of methods, we comment on the post-training code of FuseGPT for the pruning method comparison experiment. In the pure pruning method comparison experiment, FuseGPT-MI+F means that we mask the post-training code, while FuseGPT-MI implies that we additionally mask the fusion code. Implementing this method on the LLaMA-2-13b model with an NVIDIA A100-40G GPU resulted in a memory overflow, so we do not implement it.

In the layer pruning process of CoMe, we fuse two layers of the model per iteration, meaning that we reduce one layer per iteration. When pruning models from the LLaMA family, Vicuna-7b and Mistral-7b, the hyperparameter $p$ is set to 1. For the Qwen2.5-7b model, $p$ is set to 32. The Mistral-7b, Qwen2.5-7b, and LLaMA-3-8b models have high knowledge density and less redundancy in parameters, making them very sensitive to hyperparameter settings. To further mitigate performance degradation caused by merging channels with different distributions, we set a minimum parameter retention ratio $\rho$, meaning the proportion of parameters from the more critical layer cannot be less than $\rho$ during the fusion of two layers. The values of $\rho$ for the Mistral-7b, Qwen2.5-7b, and LLaMA-3-8b models are set to 0.97, 0.85, and 0.97, respectively.

Table 7: Experimental setting for post-training methods.

| Method | # Iterations | # Epochs | # Steps | Batch size | Token length |
|---|---|---|---|---|---|
| FuseGPT | 10 | 20 | 128 | 8 | 2048 |
| LLM-Streamline | 1 | 5 | 938 | 32 | 2048 |
| CoMe-mp | 7 | 1 | 2000 | 32 | 512 |
| CoMe-sp | 1 | 1 | 10000 | 32 | 512 |

## B.2 Implementation of Post-Training

Tab. 7 summarizes the post-training settings for all methods. Based on these settings, we quantify the resource consumption of each method by calculating the total number of tokens required to train a single layer, which we denote as $T_{layer}$. This metric is computed as follows:

$$T_{layer} = \# \text{ Iterations} \times \# \text{ Layers} \times \# \text{ Epochs} \times \# \text{ Steps} \times \text{Batch size} \times \text{Token length}, \quad (9)$$

where "# Layers" indicates the number of layers updated in each iteration.

The post-training process for the FuseGPT method is synchronized with the pruning process, utilizing 1,024 samples from the Wiki-2 dataset, in accordance with the settings of Pei et al. [27]. In each iteration, the parameters of one layer are merged into seven adjacent layers. Pruning ten layers requires ten iterations, with seven layers updated in each iteration. For FuseGPT, $T_{layer} \approx 2.93B$.

The LLM-Streamline trains a merged layer using 30,000 samples and employs five epochs, following the settings of Chen et al. [5]. For LLM-Streamline, $T_{layer} \approx 0.31B$.

We carry out the post-training process of CoMe after completing the pruning process. After pruning 10 layers, the pruned model has seven layers corresponding to multiple layers of the original model. Therefore, in CoMe-mp, there are seven training iterations requiring minimal training resources. CoMe-sp trains seven layers in one training round, requiring more training resources. For optimization, we utilize the AdamW optimizer with a weight decay coefficient of $1e-2$ and implement cosine decay for learning rate scheduling. The CoMe-sp employs a fixed learning rate of $1e-5$. The

CoMe-mp adopts layer-specific decaying rates during multi-layer distillation, with learning rates progressively decreasing from the shallow to the deep layers as follows: $5e-4, 2.5e-4, 1e-4, 7.5e-5, 5e-5, 2.5e-5$, and $1e-5$ for LLaMA-2-7b; $5e-4, 2.5e-4, 5e-5, 2.5e-5, 1e-5$, and $7.5e-6$ for Qwen-3-4b. For CoMe-mp, $T_{layer} \approx 0.23B$. For CoMe-sp, $T_{layer} \approx 1.15B$.

## C  Channel importance and Concatenation-based Merge

We use the channel importance for parameter division in the concatenation-based merge strategy; thus, we need to analyze the channel importance calculation for different transformer parts. Ma et al. [23] highlight that in transformer-based models, a certain correspondence exists in feature dimensions during forward propagation due to residual connections. For instance, the positional correspondence of output features from Norm, MHA, and FFN is fixed.

In the Norm part, we use the weights to scale the feature inputs. Xiong et al. [40] note that the parameters of deep layer Norms need significant enlargement to stabilize training, which is closely related to the distribution of input features. Our objective is to minimize changes in the output features of each module; therefore, we average the Norm parts of adjacent layers to maintain stability, as $\bar{\gamma} = \frac{1}{m+1} \sum_{l}^{l+m} \gamma^{(i)}$, where $m+1$ denotes the number of merge layers.

The MHA module generates three feature vectors: Query, Key, and Value. These vectors are concatenated and undergo matrix multiplication for cross-information fusion. Consequently, the weights within the heads used to generate Query, Key, and Value are tightly coupled, making it difficult to make finer divisions. Thus, we consider each head in MHA the basic unit for concatenation. We ignore the coupling between heads to further simplify the calculation of channel importance. Pruning a single head structure reduces the input dimension of the *o_project* weight (using the transformer structure in LLaMA as an example), leading to changes in the MHA output. We take the average channel importance of the reduced dimensions as the importance corresponding to each head structure.

The FFN module usually contains three weight matrices: *up_project*, *gat_project*, and *down_project*. By neglecting the coupling caused by activation functions, the information loss from channel weight pruning in *up_project* and *gat_project* maps to a reduction in intermediate feature dimensions. Therefore, we use the intermediate features and *down_project* to calculate channel importance.

## D  Posterior-based CoMe

When applying CoMe to different model architectures, it is often necessary to adjust the hyperparameter $p$ to control the parameter preservation ratio. However, the optimal ratio can vary significantly across models, which reduces the convenience and usability of CoMe. Inspired by SLEB [34] and LaCo [42], we propose an adaptive, posterior-based strategy for determining the parameter preservation ratio within CoMe, referred to as Posterior-based CoMe (CoMe-P).

CoMe-P replaces the parameter preservation ratio calculation in the layer merging process of CoMe (Eq. (5)) with a posterior-driven approach. Specifically, consider the case of merging two adjacent layers in an iteration. Let the parameter preservation ratio of the lower-indexed layer be $r$, and that of the other layer be $1-r$. We define a candidate set for $r$ as $\Gamma = \{ \frac{i}{n} \mid i = 0, 1, 2, \ldots, n \}$, where $n$ determines the granularity of the search. CoMe-P iteratively applies different preservation ratios from $\Gamma$ to generate compressed models, evaluating each candidate model on a calibration dataset using the PPL metric. The compressed model yielding the lowest PPL is selected for the final merging. The detailed algorithm of CoMe-P is presented in Alg. 2.

We set $n = 20$, with all other parameters kept consistent with the default settings of CoMe. Tabs. 8 to 10 present comparisons between CoMe-P and other methods across different models. CoMe-P achieves performance comparable to CoMe, and yields higher average accuracy and lower PPL on Qwen3-4b, Vicuna-7b, and Mistral-7B, demonstrating the effectiveness of the posterior-based approach. However, since CoMe-P is a posterior search method, the search space grows exponentially when merging more than two layers in each iteration, resulting in exponentially increased resource consumption.

# E    Analysis of CoMe-mp and CoMe-sp

To evaluate the effectiveness of CoMe-mp and CoMe-sp during the post-training process, we examine the cross-entropy loss between the student and teacher models, as shown in Fig. 12. When using CoMe-sp, the cross-entropy loss converges rapidly and stabilizes within the first 4000 steps. This phenomenon indicates an effective alignment of feature representations, as the hierarchical distillation strategy facilitates rapid convergence. In contrast, CoMe-mp shows a more linear convergence pattern, suggesting that aligning features layer by layer significantly enhances the student model's performance. However, because shallow features require processing by deeper layers, training one layer at a time results in slower convergence.

Subsection B.2 details the number of tokens used during the post-training phase. Although CoMe-sp uses fewer tokens, it requires updating

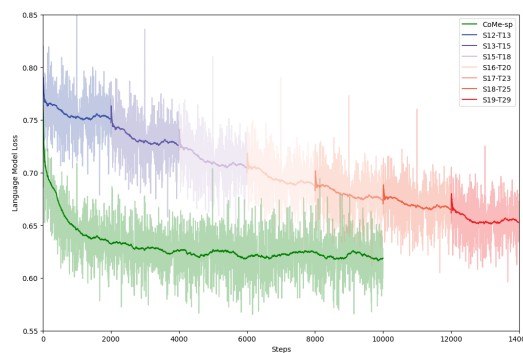

Figure 12: Cross Entropy loss curves for using CoMe-mp and CoMe-sp during post-training. The loss curves for multiple subprocesses of CoMe-mp are concatenated in the order of training.

seven times more parameters per step than CoMe-mp, necessitating greater memory resources. The overhead of memory resources is due to CoMe-sp's simultaneous optimization of multiple layers, which, while resource-intensive, allows for more efficient global information updates compared to the sequential approach of CoMe-mp.

# F    Sum-based Merge vs. Concatenation-based Merge?

**The role of Weight Sum-based Merge in both pruning and post-training processes is LESS.** Tab. 8 and Tab. 1 provide the effects of using parameter fusion (Fusion-MI+F) and only layer pruning without parameter fusion (Fusion-MI) in the FuseGPT method when pruning the LLaMA-2-7b model. In both pruning and post-training, the average accuracy score differences do not exceed 1.2 points, and PPL score differences do not exceed 0.5 points. Moreover, in pruning experiments, the performance of parameter fusion methods is worse when 20% of the parameters are pruned. It indicates that additive inter-layer parameter fusion is ineffective. In all experiments, Fusion-MI+F does not significantly improve pruning performance compared to Fusion-MI.

**The Weight Sum-based Merge does not exhibit significant differences from the DLP methods, but the Concatenation-based Merge can improve the performance of DLP methods.** In Tab. 2, after removing parameter fusion, the MKA method exhibits only a slight decrease in average accuracy scores and a slight increase in PPL, which is almost negligible. With the removal of parameter fusion, LaCo shows a slight rise in average scores and a significant decrease in PPL, indicating that the parameter fusion has a negative impact. For FuseGPT, removing parameter fusion results in a notable reduction on some datasets, such as HellaS and MMLU, a slight decrease in PPL on the C4 dataset, and a slight increase on the Wiki-2 dataset. It is difficult to conclude that parameter fusion further enhances model performance beyond layer pruning, but previous analyses suggest that FuseGPT has a minimal effect. The Weight Sum-based Merge method does not significantly differ from Direct layer pruning methods. However, when CoMe removes parameter fusion, it shows a noticeable performance decline across all test benchmarks, except for the MMLU dataset, with significant increases in PPL on the Wiki-2 and C4 datasets. It strongly indicates that Concatenation-based Merge can further enhance model performance based on DLP.

**Concatenated-based Merge is Effective, but Weight Sum-based Merge is NOT.** In Figs. 3, 11, 13 and 14, we apply both $\alpha A + (1 - \alpha)B$ (WSLP) and Concatenation-based Merge to blend parameters of two layers in varying proportions. The $\alpha$-Add method, whether merging the Self-Attention structure, the FFN structure, or the entire model layer, consistently results in a significantly increased PPL on the Wiki-2 and C4 datasets. It shows that the Weight Sum-based Merge method harms model performance, degrading performance as the fusion ratio approaches equality. Conversely, the

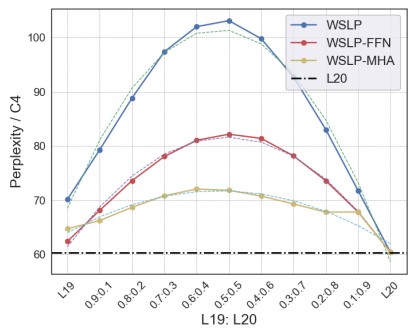

Figure 13: Merge adjacent layers with linear weight aggregation at different ratios, using the C4 calibration dataset.

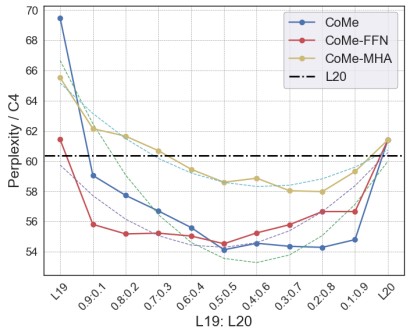

Figure 14: Merge adjacent layers with CoMe at different ratios, using the C4 calibration dataset.

Concatenation-based Merge method can reduce PPL at specific fusion ratios, preserving the model's language modeling ability.

# G  Detailed Experimental Results

In this section, we present comprehensive experimental data. The specific outcomes corresponding to Fig. 6 are detailed in Tabs. 8 to 10. Additionally, the results associated with Figs. 7 to 10 are thoroughly documented in Tabs. 11 to 14, respectively.

Table 8: The Layer Pruning Experiment on the LLaMA Family.

| Model | Sparsity | Methods | ARC-c | ARC-e | HellaS | Benchmark↑ OBQA | PIQA | WinoG | MMLU (5) | Avg↑ | RP↑ | PPL↓ C4 | Wiki-2 |
|---|---|---|---|---|---|---|---|---|---|---|---|---|---|
| LLaMA-2-7b | | Dense | 46.33 | 74.54 | 75.99 | 44.20 | 79.05 | 69.06 | 45.60 | 62.11 | 100.00 | 7.27 | 5.47 |
| | 9.0% | Mag+ | 37.97 | 66.20 | 67.65 | 39.80 | 76.44 | 59.83 | 26.79 | 53.53 | 84.56 | 9.21 | 7.01 |
| | | Taylor+ | 42.06 | 69.07 | 73.00 | 42.00 | 75.68 | 68.19 | 42.22 | 58.89 | 94.51 | 10.21 | 7.74 |
| | | ShortGPT | 43.00 | 68.77 | 71.61 | 40.40 | 76.44 | 68.67 | 45.56 | 59.21 | 95.25 | 9.33 | 7.43 |
| | | SLEB | 38.57 | 65.82 | 70.69 | 39.80 | 77.26 | 63.46 | 33.83 | 55.63 | 88.35 | 8.71 | 6.47 |
| | | FuseGPT-MI | 39.68 | 68.86 | 70.11 | 40.40 | 77.48 | 61.88 | 31.98 | 55.77 | 88.49 | 8.73 | 6.55 |
| | | FuseGPT-MI+F | 39.51 | 65.57 | 71.03 | 40.80 | 77.37 | 62.90 | 32.13 | 55.62 | 88.35 | 8.73 | 6.47 |
| | | MKA | 44.54 | 64.98 | 67.09 | 37.80 | 72.80 | 62.43 | 45.64 | 56.47 | 91.39 | 44.50 | 25.41 |
| | | LaCo | 43.43 | 68.60 | 71.78 | 40.60 | 76.39 | 68.51 | 45.39 | 59.24 | 95.35 | 9.38 | 7.46 |
| | | CoMe | 44.11 | 70.96 | 73.85 | 42.00 | 77.04 | 68.19 | 46.04 | 60.31 | 97.11 | 8.58 | 6.23 |
| | 21.0% | Mag+ | 24.32 | 44.11 | 40.23 | 31.00 | 65.72 | 53.12 | 24.58 | 40.44 | 64.10 | 37.36 | 49.17 |
| | | Taylor+ | 36.09 | 56.52 | 61.15 | 37.80 | 69.04 | 65.11 | 41.58 | 52.47 | 84.65 | 23.91 | 18.77 |
| | | ShortGPT | 36.26 | 55.85 | 62.62 | 37.20 | 70.40 | 66.30 | 39.85 | 52.64 | 84.60 | 23.31 | 18.45 |
| | | SLEB | 33.02 | 56.52 | 62.51 | 36.80 | 73.07 | 58.96 | 26.26 | 49.59 | 78.29 | 12.33 | 9.15 |
| | | FuseGPT-MI | 34.64 | 58.25 | 64.10 | 37.00 | 73.67 | 57.14 | 26.08 | 50.13 | 79.16 | 12.22 | 9.46 |
| | | FuseGPT-MI+F | 33.96 | 56.69 | 61.28 | 35.80 | 73.78 | 56.27 | 24.76 | 48.93 | 77.16 | 12.12 | 9.14 |
| | | MKA | 37.46 | 54.92 | 53.61 | 37.40 | 66.27 | 58.88 | 42.96 | 50.21 | 81.86 | 388.57 | 247.36 |
| | | LaCo | 26.79 | 49.87 | 52.69 | 33.80 | 71.55 | 55.88 | 24.77 | 45.05 | 70.90 | 18.62 | 15.85 |
| | | CoMe | 39.59 | 64.10 | 68.68 | 39.80 | 72.42 | 67.25 | 32.82 | 54.95 | 87.55 | 13.02 | 9.55 |
| | 30.0% | Mag+ | 23.98 | 39.31 | 35.77 | 27.20 | 61.75 | 51.70 | 22.96 | 37.52 | 59.49 | 52.39 | 59.73 |
| | | Taylor+ | 32.68 | 46.04 | 51.58 | 31.60 | 63.87 | 62.67 | 42.89 | 47.33 | 76.75 | 63.08 | 50.96 |
| | | ShortGPT | 31.91 | 47.39 | 45.96 | 34.80 | 63.28 | 61.48 | 38.66 | 46.21 | 75.07 | 54.92 | 49.56 |
| | | SLEB | 30.80 | 51.81 | 54.07 | 32.80 | 68.44 | 54.14 | 25.10 | 45.31 | 71.62 | 17.43 | 13.84 |
| | | FuseGPT-MI | 30.20 | 50.59 | 52.98 | 33.60 | 69.37 | 54.54 | 25.17 | 45.21 | 71.53 | 17.60 | 14.94 |
| | | FuseGPT-MI+F | 30.20 | 50.13 | 55.08 | 34.80 | 68.50 | 55.41 | 27.01 | 45.88 | 72.82 | 17.80 | 14.34 |
| | | MKA | 34.04 | 49.58 | 48.12 | 35.00 | 63.00 | 59.12 | 35.64 | 46.36 | 75.14 | 810.04 | 455.34 |
| | | LaCo | 30.97 | 49.79 | 50.14 | 35.00 | 68.34 | 53.91 | 24.84 | 44.71 | 71.11 | 39.18 | 42.67 |
| | | CoMe | 35.24 | 54.46 | 56.56 | 35.40 | 68.88 | 61.17 | 25.50 | 48.17 | 76.47 | 19.93 | 16.53 |
| | | CoMe-P | 34.64 | 54.55 | 58.35 | 35.40 | 67.95 | 61.09 | 26.89 | 48.41 | 76.89 | 18.87 | 14.74 |
| LLaMA-2-13b | | Dense | 49.15 | 77.53 | 79.39 | 45.20 | 80.52 | 72.14 | 55.16 | 65.58 | 100.00 | 6.73 | 4.89 |
| | 9.8% | Mag+ | 35.58 | 62.84 | 59.88 | 36.20 | 73.34 | 59.35 | 25.76 | 50.42 | 75.57 | 17.47 | 15.38 |
| | | Taylor+ | 45.90 | 70.71 | 76.52 | 42.20 | 78.62 | 72.30 | 43.16 | 61.34 | 92.92 | 9.57 | 7.47 |
| | | ShortGPT | 47.61 | 72.85 | 76.62 | 45.00 | 79.54 | 71.74 | 54.54 | 63.99 | 97.71 | 8.05 | 5.78 |
| | | SLEB | 42.49 | 72.31 | 74.11 | 44.00 | 79.27 | 65.51 | 42.64 | 60.05 | 91.00 | 7.81 | 5.64 |
| | | FuseGPT-MI | 40.96 | 69.02 | 74.48 | 44.00 | 79.11 | 68.82 | 42.52 | 59.84 | 90.61 | 7.83 | 5.65 |
| | | MKA | 47.01 | 69.07 | 69.42 | 45.00 | 74.81 | 65.35 | 54.02 | 60.67 | 93.31 | 34.44 | 29.88 |
| | | LaCo | 46.50 | 74.07 | 76.86 | 44.20 | 78.94 | 72.45 | 54.81 | 63.98 | 97.51 | 8.37 | 6.05 |
| | | CoMe | 47.53 | 75.25 | 78.23 | 43.20 | 79.49 | 71.98 | 55.34 | 64.43 | 98.10 | 7.61 | 5.36 |
| | 19.5% | Mag+ | 23.12 | 46.42 | 37.81 | 29.80 | 65.83 | 50.91 | 23.65 | 39.65 | 59.38 | 125.08 | 228.40 |
| | | Taylor+ | 43.26 | 65.66 | 72.09 | 40.80 | 75.30 | 70.48 | 47.24 | 59.26 | 90.09 | 13.37 | 12.12 |
| | | ShortGPT | 43.94 | 67.34 | 72.39 | 41.00 | 75.24 | 69.69 | 53.83 | 60.49 | 92.25 | 11.36 | 8.30 |
| | | SLEB | 37.88 | 64.65 | 70.59 | 42.40 | 76.82 | 64.64 | 32.32 | 55.61 | 83.83 | 9.47 | 6.85 |
| | | FuseGPT-MI | 38.91 | 63.72 | 70.97 | 40.60 | 76.93 | 67.32 | 42.57 | 57.29 | 86.66 | 9.69 | 7.04 |
| | | MKA | 40.61 | 59.60 | 57.18 | 41.40 | 68.93 | 62.90 | 53.04 | 54.81 | 84.58 | 219.41 | 206.12 |
| | | LaCo | 34.81 | 54.97 | 64.67 | 39.20 | 74.32 | 63.61 | 23.51 | 50.73 | 76.14 | 13.04 | 10.86 |
| | | CoMe | 45.14 | 72.52 | 75.87 | 42.80 | 76.50 | 70.80 | 50.35 | 62.00 | 94.30 | 9.17 | 6.29 |
| | 29.2% | Mag+ | 23.12 | 33.16 | 30.27 | 25.60 | 56.15 | 52.17 | 25.37 | 35.12 | 53.23 | 317.35 | 593.77 |
| | | Taylor+ | 38.91 | 54.97 | 62.24 | 37.20 | 70.73 | 69.61 | 48.20 | 54.55 | 83.21 | 23.96 | 28.38 |
| | | ShortGPT | 35.75 | 52.82 | 57.94 | 38.20 | 69.91 | 69.06 | 47.78 | 53.07 | 81.08 | 29.37 | 39.61 |
| | | SLEB | 34.04 | 58.59 | 63.38 | 38.60 | 75.35 | 62.35 | 26.75 | 51.29 | 76.94 | 11.64 | 8.69 |
| | | FuseGPT-MI | 37.29 | 56.90 | 64.80 | 36.60 | 74.43 | 65.04 | 30.78 | 52.26 | 78.61 | 12.65 | 9.46 |
| | | MKA | 36.95 | 53.07 | 48.67 | 36.00 | 65.56 | 60.46 | 50.72 | 50.20 | 77.39 | 759.76 | 632.28 |
| | | LaCo | 33.19 | 51.43 | 54.88 | 39.00 | 68.39 | 60.77 | 24.55 | 47.46 | 71.85 | 27.43 | 23.81 |
| | | CoMe | 42.49 | 67.05 | 69.87 | 42.60 | 73.34 | 68.98 | 51.17 | 59.36 | 90.67 | 12.64 | 8.85 |
| | | CoMe-P | 43.43 | 66.96 | 69.38 | 40.20 | 73.50 | 69.77 | 51.81 | 59.29 | 90.43 | 12.32 | 8.56 |
| LLaMA-3-8b | | Dense | 53.33 | 77.74 | 79.17 | 45.00 | 80.79 | 72.93 | 65.29 | 67.75 | 100.00 | 9.45 | 6.14 |
| | 9.2% | Mag+ | 34.73 | 64.02 | 49.37 | 36.00 | 74.16 | 54.78 | 25.35 | 48.34 | 70.80 | 25.93 | 20.44 |
| | | Taylor+ | 47.70 | 70.92 | 66.81 | 40.20 | 76.01 | 72.69 | 30.57 | 57.84 | 85.00 | 20.58 | 14.88 |
| | | ShortGPT | 47.44 | 69.99 | 73.63 | 39.80 | 76.28 | 71.43 | 63.67 | 63.18 | 92.90 | 20.08 | 15.07 |
| | | SLEB | 41.30 | 67.47 | 69.05 | 39.00 | 77.53 | 64.33 | 30.03 | 55.53 | 81.18 | 13.68 | 8.85 |
| | | FuseGPT-MI | 42.83 | 70.29 | 70.79 | 38.80 | 77.80 | 69.14 | 65.20 | 62.12 | 91.05 | 13.65 | 8.93 |
| | | FuseGPT-MI+F | 40.10 | 67.26 | 68.47 | 38.20 | 76.66 | 62.43 | 29.97 | 54.73 | 79.93 | 13.72 | 8.95 |
| | | MKA | 44.71 | 63.43 | 62.75 | 41.20 | 72.96 | 64.09 | 63.68 | 58.97 | 87.42 | 307.03 | 191.80 |
| | | CoMe | 47.70 | 72.81 | 72.28 | 40.60 | 76.50 | 74.19 | 63.65 | 63.96 | 94.08 | 14.84 | 9.52 |
| | 21.4% | Mag+ | 25.77 | 46.04 | 43.40 | 30.40 | 65.29 | 53.20 | 25.16 | 41.32 | 60.32 | 43.24 | 40.83 |
| | | Taylor+ | 31.91 | 43.60 | 35.50 | 32.20 | 60.17 | 58.80 | 33.08 | 42.18 | 62.58 | 1549.77 | 1294.94 |
| | | ShortGPT | 42.41 | 56.52 | 64.65 | 33.40 | 70.89 | 71.11 | 61.73 | 57.24 | 83.99 | 63.81 | 57.84 |
| | | SLEB | 35.75 | 58.42 | 62.29 | 34.80 | 73.83 | 57.85 | 27.43 | 50.05 | 72.99 | 18.67 | 13.38 |
| | | FuseGPT-MI | 34.56 | 59.81 | 59.03 | 34.00 | 74.37 | 56.67 | 50.43 | 52.70 | 76.98 | 19.38 | 13.44 |
| | | FuseGPT-MI+F | 33.70 | 53.91 | 61.17 | 35.80 | 72.74 | 57.93 | 26.42 | 48.81 | 71.33 | 19.03 | 13.42 |
| | | MKA | 42.58 | 60.10 | 55.90 | 40.40 | 68.88 | 62.04 | 59.27 | 55.60 | 82.66 | 1168.37 | 1004.27 |
| | | CoMe | 40.44 | 64.23 | 65.52 | 35.60 | 73.50 | 70.96 | 56.96 | 58.17 | 85.12 | 23.10 | 17.15 |
| | 30.6% | Mag+ | 22.18 | 34.22 | 33.28 | 27.00 | 57.73 | 52.49 | 24.19 | 35.87 | 52.59 | 242.47 | 254.76 |
| | | Taylor+ | 27.99 | 33.71 | 30.44 | 27.60 | 57.73 | 52.17 | 47.06 | 39.53 | 58.67 | 44214.20 | 50035.02 |
| | | ShortGPT | 30.20 | 38.13 | 32.89 | 30.20 | 59.09 | 56.75 | 41.88 | 41.31 | 61.35 | 7021.78 | 15660.69 |
| | | SLEB | 27.73 | 49.12 | 48.38 | 27.80 | 66.97 | 51.46 | 25.82 | 42.47 | 61.58 | 30.80 | 28.27 |
| | | FuseGPT-MI | 29.01 | 48.23 | 47.46 | 29.20 | 66.65 | 54.14 | 40.95 | 45.09 | 65.82 | 37.93 | 30.70 |
| | | FuseGPT-MI+F | 29.01 | 42.51 | 49.36 | 30.00 | 66.65 | 56.12 | 26.06 | 42.82 | 62.49 | 41.82 | 33.34 |
| | | MKA | 38.14 | 49.75 | 47.19 | 34.40 | 62.84 | 62.27 | 59.03 | 50.52 | 75.02 | 7447.81 | 5460.38 |
| | | CoMe | 33.70 | 50.67 | 50.42 | 31.20 | 67.08 | 60.62 | 30.99 | 46.38 | 67.86 | 48.85 | 43.56 |

Table 9: The Layer Pruning Experiment on the Vicuna-7b and Mistral-7b.

| Model | Sparsity | Methods | Benchmark↑ | | | | | | | Avg↑ | RP↑ | PPL↓ | |
| | | | ARC-c | ARC-e | HellaS | OBQA | PIQA | WinoG | MMLU (5) | | | C4 | Wiki-2 |
|---|---|---|---|---|---|---|---|---|---|---|---|---|---|
| Vicuna-7b | | Dense | 45.90 | 71.30 | 73.78 | 45.00 | 78.02 | 69.46 | 49.89 | 61.91 | 100.00 | 9.19 | 6.78 |
| | 9.0% | Mag+ | 38.40 | 64.35 | 66.04 | 40.00 | 74.21 | 59.27 | 33.11 | 53.63 | 85.59 | 11.59 | 8.54 |
| | | Taylor+ | 42.83 | 67.17 | 70.53 | 41.20 | 74.97 | 68.11 | 46.85 | 58.81 | 94.67 | 11.96 | 9.58 |
| | | ShortGPT | 43.34 | 67.68 | 70.86 | 42.00 | 75.14 | **69.85** | 49.99 | 59.84 | 96.54 | NaN | 9.13 |
| | | SLEB | 40.61 | 65.91 | 68.06 | 39.40 | 75.95 | 62.04 | 41.04 | 56.14 | 89.95 | 10.48 | 7.65 |
| | | FuseGPT-MI | 41.30 | **70.20** | 69.76 | 39.60 | **77.15** | 64.40 | 41.41 | 57.69 | 92.23 | 10.48 | 7.69 |
| | | FuseGPT-MI+F | 40.96 | 66.33 | 68.84 | 39.40 | 75.79 | 63.46 | 44.98 | 57.11 | 91.68 | 10.74 | 7.83 |
| | | MKA | **41.98** | 64.94 | 66.55 | 38.40 | 71.44 | 65.51 | **50.44** | 57.04 | 92.15 | 72.33 | 43.02 |
| | | CoMe | 43.86 | 69.53 | **72.77** | **42.80** | 75.46 | 69.06 | 49.57 | **60.44** | **97.47** | 10.48 | **7.52** |
| | 21.0% | Mag+ | 25.34 | 44.91 | 40.64 | 31.20 | 64.58 | 52.33 | 27.17 | 40.88 | 65.03 | 52.21 | 68.98 |
| | | Taylor+ | 38.65 | 58.38 | 60.71 | 35.80 | 69.10 | 65.90 | 45.58 | 53.45 | 86.10 | 21.09 | 20.45 |
| | | ShortGPT | 38.74 | 59.09 | 62.38 | 37.40 | 68.34 | 66.06 | 45.33 | 53.91 | 86.93 | 27.14 | 21.87 |
| | | SLEB | 36.26 | 61.83 | 61.66 | 35.80 | **73.72** | 59.59 | 28.31 | 51.02 | 80.84 | 13.88 | **10.52** |
| | | FuseGPT-MI | 38.57 | 62.58 | 63.03 | 37.40 | 73.23 | 59.75 | 32.37 | 52.42 | 83.59 | **13.71** | 10.55 |
| | | FuseGPT-MI+F | 36.69 | 59.72 | 61.28 | 35.00 | 73.07 | 59.35 | 29.19 | 50.61 | 80.31 | 14.78 | 10.73 |
| | | MKA | 39.85 | 54.17 | 53.34 | 38.00 | 66.97 | 61.01 | **50.61** | 51.99 | 84.95 | 540.72 | 335.40 |
| | | CoMe | **40.96** | **64.52** | **66.49** | **42.40** | 72.47 | **68.11** | 35.43 | **55.77** | **89.43** | 16.66 | 11.73 |
| | 30.0% | Mag+ | 24.15 | 40.78 | 36.13 | 27.60 | 62.30 | 50.83 | 25.06 | 38.12 | 60.48 | 72.72 | 92.79 |
| | | Taylor+ | 33.96 | 46.55 | 49.17 | 31.60 | 60.88 | 62.12 | 32.13 | 45.20 | 72.57 | 62.01 | 183.82 |
| | | ShortGPT | 33.11 | 48.40 | 48.95 | 34.60 | 64.25 | 62.90 | 41.04 | 47.61 | 76.92 | 61.91 | 59.84 |
| | | SLEB | 32.00 | 56.10 | 53.01 | 33.00 | **70.24** | 56.20 | 24.38 | 46.42 | 73.34 | NaN | NaN |
| | | FuseGPT-MI | 33.62 | 57.45 | 53.56 | 36.60 | 68.88 | 53.75 | 25.25 | 47.02 | 74.86 | 18.61 | 15.68 |
| | | FuseGPT-MI+F | 33.36 | 53.11 | 53.02 | 34.00 | 68.28 | 55.09 | 25.07 | 45.99 | 73.09 | NaN | 19.85 |
| | | MKA | 34.04 | 48.15 | 47.02 | 35.80 | 61.97 | 61.17 | **47.99** | 48.02 | 78.38 | 986.16 | 660.93 |
| | | CoMe | **36.35** | **58.84** | **56.48** | **42.40** | 68.66 | **62.98** | 25.33 | **50.15** | **80.28** | 29.55 | 18.69 |
| | | CoMe-P | 34.56 | 57.37 | 56.42 | 36.80 | 69.53 | 62.51 | 28.46 | 49.38 | 78.59 | **22.32** | **15.97** |
| Mistral-7b | | Dense | 54.01 | 79.50 | 81.06 | 44.00 | 82.05 | 74.03 | 62.52 | 68.17 | 100.00 | 8.38 | 5.25 |
| | 9.0% | Mag+ | 32.68 | 60.82 | 55.67 | 36.20 | 72.52 | 58.88 | 27.15 | 49.13 | 71.33 | 20.33 | 13.59 |
| | | Taylor+ | 44.88 | 70.83 | 75.94 | 40.60 | 79.71 | 69.69 | 52.97 | 62.09 | 90.59 | 10.13 | 6.52 |
| | | ShortGPT | 48.38 | 73.40 | 76.75 | 41.00 | 79.98 | **72.77** | **62.26** | 64.93 | **95.02** | 10.25 | 7.14 |
| | | SLEB | 43.09 | 71.09 | 74.53 | 41.40 | 79.16 | 64.64 | 41.81 | 59.39 | 86.56 | **9.76** | **6.21** |
| | | FuseGPT-MI | 45.39 | 72.10 | 74.88 | **41.60** | **80.41** | 64.80 | 41.41 | 60.08 | 87.63 | 9.80 | 6.25 |
| | | FuseGPT-MI+F | 42.49 | 70.75 | 73.49 | 41.00 | 79.76 | 66.46 | 39.62 | 59.08 | 85.98 | 9.86 | 6.31 |
| | | MKA | 43.43 | 61.03 | 53.31 | 41.20 | 67.41 | 62.75 | 58.10 | 55.32 | 82.35 | 274.34 | 203.48 |
| | | CoMe | **48.55** | **74.37** | **76.93** | 41.40 | 79.60 | 72.61 | 61.52 | **64.94** | 95.00 | 10.04 | 6.52 |
| | 21.1% | Mag+ | 23.12 | 38.47 | 33.44 | 25.60 | 59.85 | 52.09 | 23.53 | 36.59 | 53.08 | 876.24 | 1409.19 |
| | | Taylor+ | 35.24 | 54.46 | 64.30 | 33.80 | 73.72 | 61.64 | 25.05 | 49.74 | 71.87 | 19.68 | 15.34 |
| | | ShortGPT | 40.44 | 57.66 | 64.53 | 32.80 | **67.88** | **59.98** | 40.44 | 56.49 | 82.44 | 33.21 | 24.01 |
| | | SLEB | 36.95 | 61.36 | 64.81 | **39.00** | 75.24 | 61.48 | 28.83 | 52.52 | 76.44 | **13.55** | **9.25** |
| | | FuseGPT-MI | 35.07 | 58.88 | 65.71 | 37.00 | **75.68** | 55.96 | 24.98 | 50.47 | 73.13 | 14.16 | 10.00 |
| | | FuseGPT-MI+F | 34.56 | 57.20 | 65.94 | 36.60 | 74.76 | 57.30 | 25.68 | 50.29 | 72.87 | 21.73 | 15.29 |
| | | MKA | 35.84 | 42.47 | 40.42 | 33.80 | 58.49 | 57.14 | 53.98 | 46.02 | 68.75 | 36779.48 | 30245.43 |
| | | CoMe | **40.61** | **63.80** | **67.54** | 36.20 | 74.21 | 67.64 | 53.50 | **57.64** | **84.06** | 14.33 | 10.01 |
| | 30.1% | Mag+ | 25.26 | 32.07 | 30.74 | 28.20 | 54.08 | 50.28 | 25.03 | 35.09 | 51.86 | 253.79 | 288.66 |
| | | Taylor+ | 29.01 | 35.73 | 44.64 | 32.20 | 60.94 | 54.62 | 24.70 | 40.26 | 59.21 | 112.23 | 121.24 |
| | | ShortGPT | 32.00 | 29.71 | 33.56 | 31.60 | 57.51 | 56.91 | 22.72 | 37.72 | 56.16 | 760.27 | 881.51 |
| | | SLEB | 30.80 | 47.69 | 57.00 | **34.20** | 68.44 | 57.22 | 25.10 | 45.78 | 66.56 | 21.19 | 16.46 |
| | | FuseGPT-MI | 32.25 | 51.68 | 56.23 | 33.80 | 70.78 | 52.96 | 25.92 | 46.23 | 67.17 | 82.31 | 47.95 |
| | | FuseGPT-MI+F | 28.58 | 46.68 | 56.00 | 32.00 | 69.04 | 53.20 | 23.75 | 44.18 | 63.92 | 20.62 | 15.71 |
| | | MKA | 32.34 | 35.90 | 32.46 | 30.80 | 54.95 | 54.70 | 25.70 | 38.12 | 56.72 | 33065.64 | 37735.00 |
| | | CoMe | 31.48 | 51.85 | 55.81 | 31.00 | 68.77 | 58.56 | 27.47 | 46.42 | 67.10 | 22.93 | 18.32 |
| | | CoMe-P | **33.45** | **59.05** | **58.32** | 31.00 | **70.35** | **59.69** | **28.14** | **48.57** | **70.00** | **19.19** | **14.53** |

Table 10: The Layer Pruning Experiment on the Qwen-2.5-7b and Qwen-3-4b.

| Model | Sparsity | Methods | ARC-c | ARC-e | HellaS | Benchmark↑ OBQA | PIQA | WinoG | MMLU (5) | Avg↑ | RP↑ | PPL↓ C4 | Wiki-2 |
|---|---|---|---|---|---|---|---|---|---|---|---|---|---|
| Qwen-2.5-7b | | Dense | 51.11 | 77.36 | 78.95 | 47.20 | 79.65 | 73.01 | 74.16 | 68.78 | 100.00 | 11.88 | 6.85 |
| | 9.2% | Mag+ | 43.00 | 68.48 | 62.44 | 38.40 | 75.14 | 60.54 | 49.64 | 56.81 | 82.47 | 15.73 | 9.23 |
| | | Taylor+ | 47.70 | 71.84 | 67.62 | 37.00 | 73.01 | **67.01** | 65.90 | 61.44 | 88.93 | 16.96 | 11.16 |
| | | ShortGPT | 46.59 | **72.43** | 72.27 | 44.00 | 79.00 | 64.88 | 55.88 | **62.15** | **90.42** | **13.64** | **8.13** |
| | | SLEB | 46.59 | **72.43** | 72.27 | 44.00 | 79.00 | 64.88 | 55.88 | **62.15** | **90.42** | **13.64** | **8.13** |
| | | FuseGPT-MI | 43.69 | 67.26 | **72.90** | 44.00 | 78.89 | 62.90 | 63.65 | 61.90 | 89.86 | 14.42 | 8.78 |
| | | MKA | 34.64 | 47.39 | 51.55 | 35.40 | 64.15 | 58.33 | 48.95 | 48.63 | 70.82 | 25389.47 | 56126.31 |
| | | CoMe | **47.87** | 71.80 | 72.12 | **44.20** | **79.22** | 62.12 | 54.85 | 61.74 | 90.00 | 13.77 | 8.17 |
| | 21.4% | Mag+ | 29.35 | 51.52 | 50.35 | 33.40 | 67.90 | 51.85 | 28.19 | 44.65 | 64.69 | 29.12 | 16.90 |
| | | Taylor+ | 33.02 | 45.12 | 45.30 | 32.00 | 62.73 | 55.88 | **48.73** | 46.11 | 67.02 | 72.43 | 102.48 |
| | | ShortGPT | 34.98 | 62.50 | **60.41** | 37.40 | **73.94** | 54.38 | 27.87 | 50.21 | 72.84 | 17.98 | **11.37** |
| | | SLEB | 34.98 | 62.50 | **60.41** | 37.40 | **73.94** | 54.38 | 27.87 | 50.21 | 72.84 | 17.98 | **11.37** |
| | | FuseGPT-MI | 33.87 | 58.00 | 60.79 | 39.40 | 73.45 | **55.96** | 26.48 | 49.71 | 72.33 | 23.85 | 17.16 |
| | | MKA | 27.82 | 25.29 | 27.34 | 28.20 | 50.82 | 52.01 | 27.85 | 34.19 | 50.58 | 2095234.00 | 2573306.50 |
| | | CoMe | **38.82** | **65.57** | 60.38 | **40.00** | 73.61 | 55.17 | 33.10 | **52.38** | **76.36** | 24.04 | 13.81 |
| | 30.6% | Mag+ | 25.51 | 47.52 | 39.40 | 31.20 | 62.57 | 48.54 | 26.01 | 40.11 | 58.21 | 58.89 | 36.02 |
| | | Taylor+ | 25.85 | 35.90 | 31.57 | 28.60 | 58.76 | 51.30 | 25.99 | 36.85 | 53.81 | 329.33 | 422.97 |
| | | ShortGPT | 30.55 | 52.53 | 47.96 | 32.00 | 66.76 | **53.67** | 25.32 | 44.11 | 63.96 | 32.87 | 27.51 |
| | | SLEB | 27.30 | 52.57 | 48.26 | 32.00 | 68.44 | 51.78 | 26.66 | 43.86 | 63.30 | **26.33** | **17.18** |
| | | FuseGPT-MI | 26.02 | 43.81 | 43.21 | 28.80 | 66.10 | 53.43 | 25.94 | 41.04 | 59.21 | 55.32 | 51.92 |
| | | MKA | 24.91 | 25.25 | 26.29 | 29.20 | 50.71 | 49.64 | 24.21 | 32.89 | 48.69 | 14455309.00 | 17641898.00 |
| | | CoMe | **33.87** | **53.54** | **49.53** | **34.00** | **68.72** | 50.91 | 26.78 | **45.34** | **66.05** | 34.37 | 31.09 |
| Qwen-3-4b | | Dense | 51.54 | 76.43 | 73.70 | 41.20 | 77.80 | 71.03 | 73.01 | 66.39 | 100.00 | 13.31 | 7.90 |
| | 10.0% | Mag+ | 43.09 | 68.01 | 64.31 | 38.80 | **75.79** | 59.59 | 53.40 | 57.57 | 86.92 | 16.40 | 10.29 |
| | | Taylor+ | 43.17 | 63.17 | 65.15 | 34.40 | 71.65 | **68.11** | 70.55 | 59.46 | 88.99 | 21.55 | 14.88 |
| | | ShortGPT | 42.58 | 59.60 | 64.92 | 33.80 | 71.11 | 66.93 | 70.20 | 58.45 | 87.50 | 23.08 | 15.88 |
| | | SLEB | 43.43 | 67.09 | 63.37 | 39.00 | 75.30 | 60.38 | 48.83 | 56.77 | 85.91 | **15.83** | 9.69 |
| | | FuseGPT-MI | **48.63** | **73.57** | 65.54 | **39.60** | **75.79** | 62.35 | 55.63 | **60.16** | **91.01** | 16.03 | **9.43** |
| | | FuseGPT-MI+F | 48.38 | 73.53 | **65.59** | 38.80 | 75.52 | 61.96 | 55.55 | 59.90 | 90.52 | 16.03 | 9.44 |
| | | MKA | 44.11 | 63.13 | 53.44 | 36.20 | 67.19 | 59.67 | **71.19** | 56.42 | 85.20 | 497.26 | 525.95 |
| | | CoMe | 43.17 | 69.87 | 64.86 | 35.00 | 74.16 | 64.25 | 61.34 | 58.95 | 88.28 | 17.27 | 10.56 |
| | 20.1% | Mag+ | 35.49 | 61.70 | 56.05 | 37.00 | **72.42** | 54.54 | 24.98 | 48.88 | 74.22 | 24.16 | 16.60 |
| | | Taylor+ | 30.55 | 42.00 | 47.63 | 29.40 | 63.82 | 59.27 | 23.69 | 42.34 | 64.02 | 81.53 | 77.02 |
| | | ShortGPT | 36.35 | 45.12 | 52.10 | 31.00 | 65.13 | 59.75 | 38.11 | 46.79 | 70.79 | 96.83 | 130.50 |
| | | SLEB | **38.65** | **64.48** | 56.04 | **37.80** | 70.84 | 57.85 | 30.82 | 50.93 | 77.41 | 21.69 | 13.91 |
| | | FuseGPT-MI | 36.69 | 59.85 | **56.41** | **37.80** | 72.31 | 54.38 | 31.70 | 49.88 | 75.81 | **20.77** | **13.27** |
| | | FuseGPT-MI+F | 36.77 | 59.89 | 56.40 | 37.60 | 72.31 | 54.85 | 31.65 | 49.92 | 75.86 | **20.77** | **13.27** |
| | | MKA | 37.80 | 52.95 | 46.90 | 33.40 | 65.23 | **61.25** | **71.43** | **52.71** | **79.32** | 588.31 | 813.38 |
| | | CoMe | 33.19 | 56.02 | 55.29 | 30.40 | 68.55 | 59.67 | 55.10 | 51.17 | 76.30 | 30.03 | 20.75 |
| | 30.1% | Mag+ | 27.56 | 45.45 | 42.60 | **33.60** | 65.18 | 50.99 | 25.23 | 41.52 | 63.20 | 56.14 | 47.50 |
| | | Taylor+ | 28.58 | 31.23 | 32.73 | 30.60 | 54.90 | 50.20 | 23.24 | 35.93 | 55.44 | 3861.42 | 7781.82 |
| | | ShortGPT | **32.17** | 39.81 | 45.73 | 31.40 | 62.46 | 53.28 | 27.75 | 41.80 | 63.72 | 417.05 | 513.82 |
| | | SLEB | 30.97 | 53.32 | **46.42** | 31.00 | 65.67 | 53.83 | 27.16 | **44.05** | **66.50** | 39.60 | 28.11 |
| | | FuseGPT-MI | 30.63 | 55.47 | 45.90 | 30.00 | 65.61 | 52.80 | 27.33 | 43.96 | 66.17 | 33.53 | 23.49 |
| | | FuseGPT-MI+F | 30.29 | **55.64** | 45.99 | 30.40 | **65.83** | 52.25 | 27.38 | 43.97 | 66.20 | **33.52** | **23.48** |
| | | MKA | 32.08 | 38.47 | 40.95 | 30.60 | 61.53 | **60.14** | 23.43 | 41.03 | 62.61 | 2123.39 | 3208.73 |
| | | CoMe | 28.67 | 47.10 | 43.95 | 29.60 | 63.00 | 51.46 | **32.67** | 42.35 | 63.84 | 56.13 | 37.14 |
| | | CoMe-P | 27.56 | 47.73 | 43.41 | 32.00 | 63.98 | 54.06 | 29.35 | 42.58 | 64.43 | 40.83 | 28.18 |

Table 11: The detail experiment result of Fig. 7. Effect of $p$ in heuristic merge ratio.

| $p$ | ARC-c | ARC-e | HellaS | Benchmark↑ OBQA | PIQA | WinoG | MMLU (5) | Avg↑ | RP↑ | PPL↓ C4 | Wiki-2 |
|---|---|---|---|---|---|---|---|---|---|---|---|
| 1 | 35.24 | 54.46 | 56.56 | 35.40 | 68.88 | **61.17** | 25.50 | 48.17 | 76.47 | **19.93** | **16.53** |
| **2** | **35.75** | **54.84** | 56.27 | 35.40 | 68.93 | 60.77 | 25.30 | **48.18** | **76.51** | 20.70 | 16.57 |
| 4 | 34.98 | 54.76 | **57.86** | 36.40 | 67.79 | 59.75 | 24.22 | 47.97 | 76.12 | 19.87 | 17.40 |
| 8 | 34.22 | 54.17 | 56.41 | **36.60** | 68.12 | 58.48 | 26.47 | 47.78 | 76.07 | 20.94 | 19.19 |
| 16 | 32.08 | 53.62 | 56.07 | 33.60 | 68.34 | 60.62 | 26.81 | 47.31 | 74.86 | 20.30 | 17.82 |
| 32 | 30.89 | 52.78 | 55.87 | 33.40 | 68.77 | 59.43 | **26.93** | 46.87 | 74.10 | 21.43 | 18.64 |
| 64 | 32.34 | 53.28 | 56.31 | 34.60 | 67.95 | 59.91 | 26.91 | 47.33 | 75.06 | 21.69 | 18.01 |
| 128 | 32.25 | 52.48 | 56.51 | 33.20 | **69.31** | 57.38 | 27.47 | 46.94 | 74.36 | 23.29 | 18.64 |
| 256 | 32.00 | 50.80 | 55.99 | 34.80 | 67.74 | 60.22 | 27.16 | 46.96 | 74.58 | 23.97 | 19.72 |
| 512 | 31.66 | 49.33 | 55.79 | 33.20 | 67.79 | 59.04 | 26.82 | 46.23 | 73.30 | 25.03 | 20.97 |
| inf | 31.31 | 49.07 | 55.69 | 32.80 | 68.01 | 60.30 | 26.63 | 46.26 | 73.24 | 25.50 | 21.21 |

Table 12: The detail experiment result of Fig. 10. Effect of calibration data scale.

| Num | ARC-c | ARC-e | HellaS | OBQA | PIQA | WinoG | MMLU (5) | Avg↑ | RP↑ | C4 | Wiki-2 |
|---|---|---|---|---|---|---|---|---|---|---|---|
| 2 | 33.87 | 52.40 | 56.42 | 36.60 | 66.97 | 59.19 | 25.27 | 47.25 | 75.19 | 29.35 | 43.70 |
| 4 | **35.58** | 54.25 | 56.45 | **37.00** | 68.23 | 59.59 | 27.05 | **48.31** | **77.07** | 23.45 | 31.42 |
| 8 | 33.28 | 53.16 | 54.85 | **37.00** | 67.63 | 59.98 | 26.95 | 47.55 | 75.79 | 21.08 | 21.01 |
| 16 | 35.41 | 55.93 | **56.83** | 36.20 | 66.97 | 60.14 | 26.30 | 48.25 | 76.80 | 20.26 | 16.99 |
| 32 | 33.36 | 53.49 | 55.64 | 36.00 | 66.54 | 61.33 | **27.38** | 47.68 | 75.92 | 22.67 | 17.26 |
| 64 | 33.19 | 53.03 | 56.58 | 35.80 | 67.46 | 60.46 | 26.87 | 47.63 | 75.72 | 20.61 | **16.11** |
| 128 | 34.56 | 53.96 | 56.37 | 36.80 | 68.44 | **61.56** | 25.22 | 48.13 | 76.49 | 20.88 | 17.03 |
| **256** | 35.24 | **54.46** | 56.56 | 35.40 | **68.88** | 61.17 | 25.50 | 48.17 | 76.47 | 19.93 | 16.53 |
| 512 | 33.79 | 54.04 | 56.79 | 36.60 | 67.52 | 60.14 | 27.23 | 48.02 | 76.45 | **19.60** | 16.18 |

Table 13: The detail experiment result of Fig. 8. Impact of merge step granularity.

| $m$ | ARC-c | ARC-e | HellaS | OBQA | PIQA | WinoG | MMLU (5) | Avg↑ | RP↑ | C4 | Wiki-2 |
|---|---|---|---|---|---|---|---|---|---|---|---|
| **2** | **35.24** | **54.46** | **56.56** | **35.40** | **68.88** | 61.17 | 25.50 | 48.17 | 76.47 | **19.93** | **16.53** |
| 3 | 34.30 | 48.48 | 54.74 | 33.60 | 65.07 | **63.69** | **37.69** | **48.22** | **77.76** | 29.56 | 45.18 |
| 4 | 32.59 | 48.19 | 46.94 | 34.80 | 65.29 | 63.22 | 24.78 | 45.12 | 72.00 | 53.32 | 109.74 |
| 5 | 32.34 | 42.21 | 38.03 | 34.00 | 59.30 | 57.70 | 28.12 | 41.67 | 67.66 | 91.96 | 668.17 |
| 6 | 30.63 | 43.10 | 40.36 | 35.00 | 60.39 | 58.64 | 22.57 | 41.53 | 66.72 | 128.37 | 551.57 |
| 7 | 32.00 | 41.25 | 37.14 | 33.40 | 58.22 | 59.51 | 29.73 | 41.61 | 67.70 | 97.65 | 268.47 |
| 8 | 33.11 | 47.22 | 40.69 | **35.40** | 63.49 | 58.41 | 28.76 | 43.87 | 70.92 | 96.44 | 276.09 |
| 9 | 29.69 | 43.39 | 41.67 | 33.80 | 63.60 | 57.46 | 25.56 | 42.17 | 67.62 | 65.33 | 193.89 |
| 10 | 30.29 | 43.18 | 41.13 | 33.60 | 62.89 | 57.06 | 28.38 | 42.36 | 68.27 | 78.85 | 303.17 |
| 11 | 30.46 | 41.58 | 40.24 | 34.20 | 61.37 | 59.91 | 24.71 | 41.78 | 67.20 | 104.82 | 470.91 |

Table 14: The detail experiment result of Fig. 9. Impact of calibration datasets.

| Dataset | ARC-c | ARC-e | HellaS | OBQA | PIQA | WinoG | MMLU (5) | Avg↑ | RP↑ | C4 | Wiki-2 |
|---|---|---|---|---|---|---|---|---|---|---|---|
| wiki2 | **35.24** | **54.46** | 56.56 | 35.40 | 68.88 | 61.17 | 25.50 | 48.17 | 76.47 | 19.93 | **16.53** |
| C4 | 34.30 | 52.95 | **57.25** | **35.60** | **69.80** | 62.04 | 26.39 | **48.33** | 76.71 | **19.32** | 21.05 |
| PG19 (2) | 34.39 | 50.21 | 53.74 | 34.00 | 67.25 | 60.69 | 27.90 | 46.88 | 74.77 | 21.28 | 21.15 |
| MMLU | 33.70 | 50.80 | 54.13 | 33.20 | 66.00 | **63.77** | 27.20 | 46.97 | 74.67 | 28.12 | 35.79 |
| Aplaca | 34.39 | 51.77 | 53.54 | 35.00 | 67.25 | **63.77** | **29.87** | 47.94 | 76.61 | 25.65 | 30.57 |

---

**Algorithm 1** Progressive Concatenation-based Layer Merging Strategy (CoMe)

---

**Input:** calibration dataset $\mathcal{D}$, original model $M$, the number of layers skipped in SBI $m$, skewness exponent $p$, target layer number $L$, minimum retention ratio $\rho \in (0, 1)$

**Output:** Pruned model $M'$, layer mapping $\mathcal{P} = [\{a_1, b_1\}, ..., \{a_N, b_N\}]$

 1: Initialize $\mathcal{P} \leftarrow \emptyset$, $M' \leftarrow M$
 2: **while** NUMLAYERS$(M') > L$ **do**
 3:     $m' \leftarrow \min(m, \text{NUMLAYERS}(M') - L)$
 4:     $\{S^{(l)}\} \leftarrow$ COMPUTECHANNELSENSITIVITY$(M', \mathcal{D})$
 5:     $\{BI_l\} \leftarrow$ COMPUTEBISCORES$(\{\mathbf{H}^{(l)}\})$
 6:     $\{SBI_{l:l+m'}\} \leftarrow$ COMPUTESBISCORES$(\{\mathbf{H}^{(l)}\}, m')$
 7:     $(l^*, l^* + m') \leftarrow \arg\min SBI_{l:l+m}$
 8:     $\{r_t\} \leftarrow$ ADJUSTRETENTIONRATIOS$(\{BI_t\}_{t=l^*}^{l^*+m'}, p, \rho)$
 9:     $W^{(\text{merge})} \leftarrow$ CONCATENATIONBASEDLAYERMERGE$(\{W^{(t)}, S^{(t)}, r_t\}_{t=l^*}^{l^*+m'})$
10:     Replace layers $[l^*, ..., l^* + m']$ with $W^{(\text{merge})}$ in $M'$
11:     $\mathcal{P} \leftarrow \mathcal{P} \cup [\{l^* + m', \text{new layer index in } M'\}]$
12: **end while**
13: **return** $M', \mathcal{P}$
14:
15: **function** COMPUTECHANNELSENSITIVITY$(M', \mathcal{D})$
16:     **for** each layer $l \in M'$ **do**
17:         $\mathbf{H}^{(l)} \leftarrow$ FORWARDPASS$(M', \mathcal{D}, l)$
18:         $S^{(l)} \leftarrow \{\mathbb{E}_{\mathcal{D}}[|x_i| \sum_k |w_{i,k}|]\}_{i=1}^{u_l}$               ▷ Eq. (1)
19:     **end for**
20:     **return** $\{S^{(l)}\}_{l=1}^{L}$
21: **end function**
22: **function** COMPUTEBISCORES$(\{\mathbf{H}^{(l)}\})$
23:     **for** $l = 1$ to NUMLAYERS$(M')$ **do**
24:         $BI_l \leftarrow 1 - \mathbb{E}_{\mathcal{D}}\left[\frac{\mathbf{H}^{(l-1)\top}\mathbf{H}^{(l)}}{\|\mathbf{H}^{(l-1)}\|_2\|\mathbf{H}^{(l)}\|_2}\right]$               ▷ Eq. (2)
25:     **end for**
26:     **return** $\{BI_l\}$
27: **end function**
28: **function** COMPUTESBISCORES$(\{\mathbf{H}^{(l)}\}, m')$
29:     **for** $l = 1$ to NUMLAYERS$(M') - m'$ **do**
30:         $SBI_{l:l+m'} \leftarrow 1 - \mathbb{E}_{\mathcal{D}}\left[\frac{\mathbf{H}^{(l-1)\top}\mathbf{H}^{(l+m')}}{\|\mathbf{H}^{(l-1)}\|_2\|\mathbf{H}^{(l+m')}\|_2}\right]$               ▷ Eq. (3)
31:     **end for**
32:     **return** $\{SBI_{l:l+m'}\}$
33: **end function**
34: **function** ADJUSTRETENTIONRATIOS$(\{BI_t\}_{t=l^*}^{l^*+m'}, p, \rho)$
35:     $r_t \leftarrow BI_t^p / \sum_i BI_i^p$ for $t \in [l^*, l^* + m']$               ▷ Eq. (5)
36:     **if** $\max r_t < \rho$ **then**
37:         $r_{\arg\max BI_t} \leftarrow \rho$
38:         $t^* \leftarrow \arg\max BI_t$
39:         $\sum' \leftarrow \sum_{t \neq t^*} r_t$
40:         $r_t \leftarrow (1-\rho)r_t / \sum'$ for $t \neq t^*$
41:     **end if**
42:     **return** $\{r_t\}$ normalized to $\sum r_t = 1$
43: **end function**
44: **function** CONCATENATIONBASEDLAYERMERGE$(\{W^{(t)}, S^{(t)}, r_t\}_{t=l^*}^{l^*+m'})$
45:     **for** each layer $t \in [l^*, l^* + m']$ **do**
46:         $k_t \leftarrow r_t \times |S^{(t)}|$
47:         $\mathcal{T}_t \leftarrow$ Top-$k_t$ indices sorted by $S^{(t)}$
48:     **end for**
49:     $W^{(\text{merge})} \leftarrow \bigoplus_{t=l^*}^{l^*+m'} W^{(t)}[:, \mathcal{T}_t]$               ▷ Eq. (4)
50:     **return** $W^{(\text{merge})}$
51: **end function**

---

**Algorithm 2** Progressive Posterior-based CoMe (CoMe-P)

---

**Input:** Calibration dataset $\mathcal{D}$, original model $M$, target layers number $L$, search granularity $n$
**Output:** Pruned model $M'$, layer mapping $\mathcal{P} = [\{a_1, b_1\}, ..., \{a_N, b_N\}]$
 1: Initialize $\mathcal{P} \leftarrow \emptyset$, $M' \leftarrow M$,
 2: Generate parameter preservation ratio candidate set $\Gamma = \{\frac{i}{n} \mid i = 0, 1, 2, \ldots, n\}$
 3: **while** NUMLAYERS$(M') > L$ **do**
 4:     $\{S^{(l)}\} \leftarrow$ COMPUTECHANNELSENSITIVITY$(M', \mathcal{D})$
 5:     $\{SBI_{l:l+1}\} \leftarrow$ COMPUTESBISCORES$(\{\mathbf{H}^{(l)}\}, 1)$
 6:     $(l^*, l^* + 1) \leftarrow \arg\min SBI_{l:l+1}$
 7:     **for** each $r$ **in** $\Gamma$ **do**
 8:         $W^{(\text{merge})} \leftarrow$ CONCATENATIONBASEDLAYERMERGE$(\{W^{(t)}, S^{(t)}, r_t\}_{t=l^*}^{l^*+1})$
 9:         $M'' \leftarrow$ Replace layers $[l^*, l^* + 1]$ with $W^{(\text{merge})}$ in $M'$
10:         $ppl \leftarrow PPL(M'')$
11:     **end for**
12:     $M' \leftarrow$ The $M''$ has the lower $ppl$
13:     $\mathcal{P} \leftarrow \mathcal{P} \cup [\{l^* + 1, \text{new layer index in } M'\}]$
14: **end while**
15: **return** $M', \mathcal{P}$

---

**Algorithm 3** CoMe Single-Process Post-training (CoMe-sp)

---

**Input:** training data $\mathcal{D}_{\text{train}}$, layer mapping $\mathcal{P} = [\{a_1, b_1\}, ..., \{a_N, b_N\}]$, teacher model $M$, student model $M'$, learning rate $\eta$, optimizer $\Omega$, batch size $B$
**Output:** Optimized student model $M'$
 1: Initialize $\Omega \leftarrow$ ADAM$(\{\theta_{b_i} \mid \{a_i, b_i\} \in \mathcal{P}\}, \eta)$         $\triangleright$ Optimize merged layers only
 2: **for** epoch $= 1$ to $E_{\text{global}}$ **do**
 3:     **for** $\mathcal{B} \leftarrow$ BATCHLOADER$(\mathcal{D}_{\text{train}}, B)$ **do**
 4:         $\{\mathbf{H}^{(t, a_i)}\}_{i=1}^N \leftarrow$ GETACTIVATIONS$(M, x, \{a_i \mid \{a_i, b_i\} \in \mathcal{P}\})$
 5:         $\{\mathbf{H}^{(s, b_i)}\}_{i=1}^N \leftarrow$ GETACTIVATIONS$(M', x, \{b_i \mid \{a_i, b_i\} \in \mathcal{P}\})$
 6:         **return** $\{\mathbf{H}^{(t, a_i)}, \mathbf{H}^{(s, b_i)}\}_{i=1}^N$
 7:         $\mathcal{L}_{\text{total}} \leftarrow 0$
 8:         **for** $i = 1$ **to** $N$ **do**
 9:             $\mathcal{L}_{\text{KL}}^{(i)} \leftarrow \frac{1}{N} D_{\text{KL}}(\sigma(\mathbf{H}^{(t, a_i)}) \parallel \sigma(\mathbf{H}^{(s, b_i)}))$     $\triangleright$ Eq. (6)
10:             $\mathcal{L}_{\text{total}} \leftarrow \mathcal{L}_{\text{total}} + \mathcal{L}_{\text{KL}}^{(i)}$     $\triangleright$ Eq. (8)
11:         **end for**
12:         $\Omega.zero\_grad()$
13:         $\mathcal{L}_{\text{total}}.\text{BACKWARD}()$
14:         $\Omega.step()$
15:     **end for**
16: **end for**

---

---

**Algorithm 4** CoME Multi-Process Post-training (CoMe-mp)

---

**Input:** training data $\mathcal{D}_{\text{train}}$, layer mapping $\mathcal{P} = [\{a_1, b_1\}, ..., \{a_N, b_N\}]$, teacher model $M$, student model $M'$, learning rate $\{\eta_1, ..., \eta_{|\mathcal{P}|}\}$, optimizer $\Omega$, batch size $B$

1: **for** $k = 1$ **to** $|\mathcal{P}|$ **do**                                      ▷ Layerwise progression
2:     $\{a_k, b_k\} \leftarrow \mathcal{P}[k]$
3:     $\Omega_k \leftarrow \text{ADAM}(\theta_{b_k}, \eta_k = \eta_k)$
4:     **for** epoch $= 1$ **to** $E_{\text{local}}$ **do**
5:         **for** $\mathcal{B} \leftarrow \text{BATCHLOADER}(\mathcal{D}_{\text{train}}, B)$ **do**
6:             $\mathbf{H}^{(t,a_k)} \leftarrow \text{GETSINGLEACTIVATION}(M, x, a_k)$
7:             $\mathbf{H}^{(s,b_k)} \leftarrow \text{FORWARDTOLAYER}(M', x, b_k)$
8:             $\mathcal{L}_{\text{KL}} \leftarrow D_{\text{KL}}(\sigma(\mathbf{H}^{(t,a_k)}) \| \sigma(\mathbf{H}^{(s,b_k)}))$          ▷ Eq. (6)
9:             $\Omega_k.zero\_grad()$
10:            $\mathcal{L}_{\text{KL}}.\text{BACKWARD}()$
11:            $\Omega_k.step()$
12:        **end for**
13:    **end for**
14: **end for**
15: **return** $M'$

---

