# OpenReview forum: "Layer as Puzzle Pieces: Compressing Large Language Models through Layer Concatenation"
_NeurIPS.cc/2025/Conference — NeurIPS 2025 poster_

### Official Review · Reviewer_1RXD · 2025-06-30

**Clarity:** 3
**Significance:** 3
**Originality:** 2
**Rating:** 3
**Confidence:** 4

**Summary:**

This paper introduces CoMe, a structured compression framework for large language models that integrates: a channel sensitivity metric that utilizes activation intensity and weight norms for fine-grained channel selection, a concatenation-based layer merging method and a hierarchical distillation protocol. Extensive experiments on multiple LLMs (e.g. LLaMA‑2‑7b, Qwen‑3‑4b) and three sparsity levels (10/20/30%) demonstrate that CoMe consistently outperforms existing layer‑pruning baselines.

**Questions:**

Please elaborate on the difference between CoMe and Laco [1].

[1] Yang, Yifei, Zouying Cao, and Hai Zhao. "Laco: Large language model pruning via layer collapse." arXiv preprint arXiv:2402.11187 (2024).

**Ethical Concerns:**

["NO or VERY MINOR ethics concerns only"]

**Final Justification:**

The current manuscript has deficiencies in experimental comparisons, which make the experimental conclusions less credible and fair, so I maintain the original score. The main issues are as follows:

- The computational cost comparison is not uniform (unfair): The table sets the sequence length of FuseGPT/LLM-Streamline to 2048, while that of CoMe is 512. Direct comparison under such non-uniform input lengths is unfair.
- Lack of comparison with LoRA: The paper uses Hierarchical Distillation for fine-tuning and accuracy recovery, but does not compare with LoRA (a representative method for efficient parameter fine-tuning). LoRA is often used as a strong baseline for resource-constrained fine-tuning.

**Limitations:**

yes

**Paper Formatting Concerns:**

No Paper Formatting Concerns

**Quality:**

3

**Strengths And Weaknesses:**

- Strengths
  - This paper is easy to follow.
  - The ablation experiments are very detailed.
  - Progressive Concatenation-based Layer Merge: Instead of simply linearly weighting or averaging layer weights, authors fuse high-value channels of adjacent layers according to sensitivity to construct an optimally aligned subspace, thereby maximizing the representation capability of the original model under the same parameter budget.
- Weaknesses
  - Only the results of the 7B system model are provided. The conclusions may not be transferable to other larger-sized models (e.g. LLAMA3-70B).
  - There is a lack of theoretical evidence to prove why merging is better than direct pruning.
  - The paper does not conduct a quantitative comparative analysis between the proposed knowledge distillation method and traditional LoRA fine-tuning in terms of downstream task performance, training time, number of parameters, and computing resource consumption, which makes it impossible for readers to evaluate the effectiveness of hierarchical distillation protocol.

---

> ### Author Rebuttal · Authors · 2025-07-30
>
> Thank you for taking the time to review our manuscript. We greatly appreciate your valuable feedback. Below, we provide our responses to your comments:
> ## **[Question 1] Difference between CoMe and LaCo**
> >LaCo belongs to the category of WSLP methods, while CoMe differs fundamentally from LaCo in terms of layer selection, fusion strategy, and post-training protocol.
>
> > * **Fusion Strategy**:
> CoMe employs a concatenation-based merging at the sub-module level, whereas LaCo essentially utilizes a weighted sum of layer weights.
>
> > * **Layer Group Selection**:
> CoMe uses the SBI score to select a group of adjacent layers for merging, requiring only a single forward pass on a calibration dataset in one iteration. In contrast, LaCo sets a threshold $\mathcal{T}$ and traverses from shallow layer index $\mathcal{L}$ to deep layer index $\mathcal{H}$, merging $\mathcal{I}$ layers at a time. If the similarity between the pruning model’s output and the original model’s output is within a threshold $\tau$, the fused layer will be adopted. Otherwise, it rejects the merging process and retries with different layers until the target number of layers is reached or all layers are traversed. LaCo is a post-hoc pruning method and, due to multiple hyperparameters, it is challenging to obtain a model with the required number of layers. Therefore, we only reproduced LaCo experiments on the LLaMA-2 series models.
>
> >* **Post-training**:
> CoMe employs a fine-grained, layer-wise post-training protocol, which operates by aligning the output distributions of individual layers during the training process. In contrast, LaCo adopts a model-wise post-training approach that aligns the logits of the original model and the pruned model through conventional knowledge distillation. Due to our resource constraints (2 × A100 40G), we only compared CoMe with layer-wise post-training methods. We were unable to reproduce LaCo’s post-training results and thus excluded it from the comparative analysis.
>
> ## **[Weakness 2] Theoretical Evidence for Merging**
> >The theoretical motivation for CoMe is grounded in two key observations from previous layer pruning studies, as detailed in Section 3 of our manuscript. First, DLP directly prunes entire layers, which can result in the loss of important weights. Second, WSLP ignores the correlation between the weights of adjacent layers, and its blind weighted fusion often leads to over-smoothed weight distributions. Based on these insights, we propose a novel layer pruning framework. CoMe decomposes each layer into submodules whose inputs and outputs are independent, so that removing or adding submodules does not affect the input or output of the remaining submodules. For example, as analyzed in Appendix  C, channels in the FFN structure and heads in the MHA structure can be treated as such submodules. By concatenating these submodules, CoMe avoids weight smoothing and more effectively preserves important weights or functional components.
>
> >It is important to note that layer pruning is inherently a lossy process, and there is currently no effective theoretical method to quantify the extent of this loss. This limitation is common across many layer pruning approaches (e.g., LaCo, MKA, SLEB, ShortGPT, etc.), making the field largely empirical.
>
> >In our manuscript, we provide detailed experimental comparisons between DLP and WSLP (see Figure 3 and Figure 13), as well as between DLP and CoMe (see Figure 10 and Figure 14), clearly illustrating the effects of direct layer pruning, weighted sum fusion, and concatenation-based fusion on model performance. Ablation studies of the fusion operation in Table 2 further demonstrate that CoMe is more effective than WSLP.
>
> ## **[Weakness 1] Experiment on Larger Models**
> >Due to limited computational resources (2 × A100 40G), we were unable to conduct experiments on models larger than 20B parameters. We have provided experimental results on models of various scales, including 4B, 7B, 8B, and 13B. We plan to extend our experiments to 70B and even larger models when resources permit in the future.
>
> ## **[Weakness 3] Lack of Quantitative Comparison with LoRA Fine-tuning**
> >We appreciate the reviewer’s suggestion. Our method is orthogonal to LoRA fine-tuning, as the two approaches serve fundamentally different purposes. LoRA aims to reduce the number of trainable parameters through low-rank decomposition for parameter-efficient fine-tuning. CoMe’s post-training protocol is designed to rapidly align the merged layers using information obtained during pruning, thereby restoring model performance.
> Additionally, because CoMe’s pruning process is dynamic, the number of layers and training time required for post-training also vary accordingly.
> In our paper, we compared post-training methods that are more closely related to CoMe’s motivation (Table 1) and provided comparisons of training processes and resource consumption (Appendix B.2 and Table 7).
>
> We hope these responses can address your concerns. If you have further questions, we would be happy to provide additional clarification.

---

> > ### Comment · Reviewer_1RXD · 2025-08-03
> >
> > First, thank the authors for their detailed response and additional experiments. However, I carefully review Table 1 in the authors' response and Table 7 in the Appendix and found no experiments comparing training time, number of parameters, and computing resource consumption with traditional LoRA fine-tuning. This makes it impossible to evaluate the effectiveness of the hierarchical distillation protocol.

---

> > > ### Author Response · Authors · 2025-08-03
> > > **Clarification on the Absence of Comparisons with LoRA**
> > >
> > > We sincerely appreciate your positive feedback.
> > >
> > > # Below, we clarify why we did not include comparisons with parameter-efficient fine-tuning methods such as LoRA.
> > >
> > > >## 1. Different Motivation
> > > >Parameter-efficient fine-tuning is an active area of research focused on rapidly adapting pre-trained models to downstream tasks, with LoRA being one of the most prominent methods. LoRA reduces the number of trainable parameters through low-rank decomposition. However, it is essential to emphasize that **post-training methods designed for layer pruning, such as CoMe, LLM-Streamline [1], and FuseGPT [2], are primarily intended to efficiently restore the performance of pruned models to a level comparable to the original model.** In the literature, state-of-the-art post-training methods for pruning, including LLM-Streamline [1] and FuseGPT [2], do not include direct comparisons with LoRA, as these approaches serve fundamentally different purposes.
> > >
> > > >## 2. Different Loss Functions
> > > >CoMe, LLM-Streamline [1], and FuseGPT [2] utilize **layer-wise distillation losses** to align intermediate features between teacher and student models, which is suitable only for feature alignment in the context of pruning. In contrast, LoRA typically employs **model-wise loss functions** based on logits alignment, which has broader applicability and can be used for both direct fine-tuning and knowledge distillation.
> > >
> > > >## 3. Differences in Fine-tuned Weight Matrices
> > > >LoRA is capable of fine-tuning arbitrary weight matrices within the model. In contrast, post-training methods designed for layer pruning generally only fine-tune those weight matrices that have been modified during pruning. Furthermore, the indices and number of layers that are fine-tuned during post-training vary significantly across CoMe, LLM-Streamline [1], FuseGPT [2], and similar methods. Even within a single method, such as CoMe or FuseGPT, the specific layers and total number of layers requiring fine-tuning depend on the model architecture and the pruning configuration. As the set of pruned layers changes, so too does the set of layers that require fine-tuning. Therefore, **it is neither practical nor meaningful to compare the efficiency of CoMe directly, LLM-Streamline [1], or FuseGPT [2] with LoRA in terms of training time, parameter count, or resource usage.**
> > >
> > > # Our Experimental Comparisons
> > >
> > > >In our manuscript, **we compare CoMe’s post-training protocol with other state-of-the-art post-training methods (LLM-Streamline [1] and FuseGPT [2]) that are specifically tailored for layer pruning (see Table 1 and Table 7).**
> > >
> > > If you have any further questions, we would be happy to discuss them in more detail.
> > >
> > > *[1] Chen, Xiaodong, et al. “Streamlining Redundant Layers to Compress Large Language Models.” Proceedings of the Thirteenth International Conference on Learning Representations (ICLR 2025), 2025.*
> > >
> > > *[2] Pei, Zehua, et al. “FuseGPT: Learnable Layers Fusion of Generative Pre-trained Transformers.” arXiv preprint arXiv:2411.14507 (2024).*

---

> > > > ### Author Response · Authors · 2025-08-06
> > > > **Addition of an Explanation for CoMe Computational Cost Comparison**
> > > >
> > > > To facilitate a more comprehensive comparison between CoMe and other state-of-the-art post-training methods tailored for layer pruning (including LLM-Streamline and FuseGPT), we present the following analysis.
> > > >
> > > > ## Comparison of Post-Training Computational Costs for Pruning Methods
> > > >
> > > > >**The computational cost of post-training strategies specifically designed for pruning methods often closely relates to the pruning approach itself, and due to variations in implementation, it is challenging to make a completely fair comparison. Below, we present a comparison of the computational costs of four post-training methods under a fixed setting.**
> > > >
> > > > >CoMe, LLM-Streamline, and FuseGPT require forward passes to compute hidden features for layer-wise fine-tuning. The cost depends on which layers are pruned, as fine-tuning deeper layers requires more resources to obtain their inputs and outputs. For this analysis, we assume all four methods incur similar costs for this step and **focus solely on the computational cost associated with weight fine-tuning**.
> > > >
> > > > >**CoMe-mp, FuseGPT, and LLM-Streamline fine-tune one layer per iteration. Therefore, we use the cumulative number of tokens processed during fine-tuning as a proxy for computational cost**. CoMe-sp fine-tunes multiple layers simultaneously, and since not all fine-tuned layers are necessarily adjacent, some non-fused layers may also participate in the forward pass, resulting in slightly higher computational costs than those reported below.
> > > >
> > > > >We report the total number of tokens processed during the post-training phase when pruning 10 layers from the LLaMA2-7b model. ‘# Steps’ denotes the number of steps per epoch, and ‘# Layers’ indicates the number of layers updated per iteration.
> > > >
> > > > * *LLM-Streamline*: Fine-tunes one layer per iteration for 5 epochs with 938 steps per epoch. Thus, the cumulative number of tokens is:
> > > >
> > > > $1 \text{(\\# Iterations)} \times 1 \text{(\\# Layers)} \times 5 \text{(\\# Epoch)} \times 938 \text{(\\# Steps)} \times 32 \text{(\\# Batch size)} \times 2048 \text{(\\# Sequence Length)} \approx 0.31 \text{B} $
> > > >
> > > > * *FuseGPT*: When pruning 10 layers, performs 10 fusion iterations, updating 7 layers in an iteration, across 20 epochs with 128 steps per epoch. The cumulative number of tokens is:
> > > >
> > > > $10 \text{(\\# Iterations)} \times 7 \text{(\\# Layers)} \times 20 \text{(\\# Epoch)} \times 128 \text{(\\# Steps)} \times 8 \text{(\\# Batch size)} \times 2048 \text{(\\# Sequence Length)} \approx 2.93 \text{B} $
> > > >
> > > > * *CoMe-mp*: After pruning, 7 layers are updated, with one layer per iteration, over one epoch with 2000 steps per epoch:
> > > >
> > > > $7 \text{(\\# Iterations)} \times 1 \text{(\\# Layers)} \times 1 \text{(\\# Epoch)} \times 2000 \text{(\\# Steps)} \times 32 \text{(\\# Batch size)} \times 512 \text{(\\# Sequence Length)} \approx 0.23 \text{B} $
> > > >
> > > > * *CoMe-sp*: Fine-tunes 7 layers in a single iteration, over one epoch with 10,000 steps:
> > > >
> > > > $1 \text{(\\# Iterations)} \times 7 \text{(\\# Layers)} \times 1 \text{(\\# Epoch)} \times 10,000 \text{(\\# Steps)} \times 32 \text{(\\# Batch size)} \times 512 \text{(\\# Sequence Length)} \approx 1.15 \text{B} $
> > > >
> > > > >In practice, the number of layers requiring updates after pruning 10 layers with CoMe may range from 1 to 10, leading to different levels of resource consumption.
> > > >
> > > > >From these calculations, the order of computational cost is: FuseGPT > CoMe-sp > LLM-Streamline > CoMe-mp. As shown in Table 1, the ranking of model performance is: CoMe-mp > CoMe-sp > LLM-Streamline > FuseGPT. **CoMe-mp achieves the best benchmark performance with the lowest post-training cost, outperforming both FuseGPT and LLM-Streamline. Although CoMe-sp requires more computational resources, its post-training performance is significantly better than that of the other three methods**.

---

> > > > > ### Comment · Reviewer_1RXD · 2025-08-09
> > > > >
> > > > > About the Comparison of Post-Training Computational Costs for Pruning Methods. As the author mentioned, the sequence length of FuseGPT and LLM-Streamline is $2048$, while the CoMe is only $512$, this is very **unfair**. FuseGPT and LLM-Streamline can also use $512$ to conduct experiments. If doing that, the computational cost would be lower than CoMe.
> > > > >
> > > > > Besides, comparing with LoRA is necessary, since you introduce a Hierarchical Distillation Strategy to perform fine-tuning. LoRA can also be used to perform fine-tuning.

---

> > > > > > ### Author Response · Authors · 2025-08-09
> > > > > > **Clarification on Experimental Fairness and the Scope of Our Method**
> > > > > >
> > > > > > We appreciate your thoughtful feedback and suggestions. We would like to take this opportunity to clarify several points raised in your review.
> > > > > >
> > > > > > ## Comparison between LoRA and the Hierarchical Distillation Strategy:
> > > > > >
> > > > > > >LoRA is designed to **reduce the rank of parameter matrices**. The hierarchical distillation strategy is designed to **select which layers to fine-tune**.
> > > > > > LoRA is primarily designed for **parameter-efficient** fine-tuning, aiming to **reduce the number of trainable parameters** during the training process. In contrast, the hierarchical distillation strategy is a **strategy** specifically designed for the post-training stage of layer pruning, aiming to **utilize the layer mapping between the original and pruned models, obtained during pruning, in the distillation process**.
> > > > > >
> > > > > > >When applying LoRA for fine-tuning, although only a subset of parameters are updated, **the entire model still needs to be loaded into memory** during training. In contrast, in the hierarchical distillation framework, only **the specific layer being fine-tuned needs to be stored in memory**, resulting in significantly reduced memory consumption. *If LoRA were applied to a single layer, the comparison would shift to evaluating the effectiveness of hierarchical single-layer full-parameter fine-tuning versus hierarchical single-layer LoRA fine-tuning, which is beyond the scope of this work.*
> > > > > >
> > > > > > ## Regarding the Number of Fine-Tuning Tokens:
> > > > > >
> > > > > > >LLM-Streamline, FuseGPT, and CoMe adopt distinct fine-tuning strategies and implementations, making it challenging to directly and fairly compare the computational costs of their fine-tuning processes. Therefore, **we use the number of tokens as an approximate metric for resource consumption, aiming to demonstrate that our method is effective under low-resource constraints, rather than claiming absolute efficiency superiority**. We chose a sequence length of 512 to further address scenarios with limited computational resources. Independent of performance considerations, shorter sequence lengths reduce memory usage during computation and also decrease the wall-clock time per training step compared to a sequence length of 2048. Specifically, the computational complexity for the MHA component is $O(L^2)$ and for the FFN component is $O(L)$, where $L$ is the sequence length. Theoretically, the wall-clock time per step for **a sequence length of 2048 is approximately 8-12 times greater than that for 512**. However, due to technical limitations, it is difficult to provide highly optimized implementations for each method, making it challenging to fairly compare the exact resource consumption of different hierarchical post-training strategies.
> > > > > >
> > > > > > **The main contribution of our work is the proposal of a novel and effective layer pruning method, along with an efficient post-training strategy that is well-matched to the pruning process. Due to the varying implementations of different methods, we do not claim that our approach is the most resource-efficient among all existing hierarchical post-training methods. Instead, our main objective is to demonstrate the effectiveness and practicality of our proposed pruning and post-training framework.**

---

> > > > > > > ### Comment · Reviewer_1RXD · 2025-08-09
> > > > > > >
> > > > > > > Thank you to the authors for the response and the additional results. I believe my score matches the paper. Therefore, I have decided to maintain my original score.

---

> ### Author Response · Authors · 2025-08-08
> **Request for Further Comments**
>
> Thank you very much for your detailed review and valuable feedback on our manuscript. We truly appreciate the time and effort you dedicated to evaluating our work. We highly value your assessment and kindly request that you let us know if you have any remaining questions or points that require further clarification. Your additional feedback would be immensely helpful in helping us improve the quality of our work. Thanks again for your time and consideration.

---

### Official Review · Reviewer_8TsA · 2025-07-01

**Clarity:** 3
**Significance:** 2
**Originality:** 3
**Rating:** 4
**Confidence:** 3

**Summary:**

This paper proposes CoMe, a structured pruning framework for large language models that improves upon existing methods by avoiding performance loss from direct layer removal and the limitations of linear layer fusion. It introduces a channel sensitivity metric, a concatenation-based merging strategy, and a hierarchical distillation approach to preserve critical information and support effective post-training recovery. Experiments across multiple models and benchmarks demonstrate consistent performance gains over prior work.

**Questions:**

- The evaluation focuses exclusively on multiple-choice benchmarks such as ARC, PIQA, and MMLU. Could the authors clarify whether the proposed compression method has been tested on open-ended generation tasks or real-world downstream applications?

-  Could the authors clarify why this particular formulation based on expected output perturbation was chosen over more established alternatives such as gradient-based or second-order importance metrics? Additionally, has the metric been evaluated in isolation across diverse settings to assess its general applicability beyond the CoMe framework?

- The channel retention ratio p seems to be set empirically. Please explain the selection principle of p and the adaptive strategy at different layers and different model sizes. If p is not properly set, how much performance will be degraded?

**Ethical Concerns:**

["NO or VERY MINOR ethics concerns only"]

**Final Justification:**

The author's explanation effectively addresses my concern regarding the diversity of benchmarks and the evaluation scope in Layer Pruning. I find this clarification convincing and therefore have decided to raise my score.

However, I believe that Weakness 3 still requires additional experimental support. The current response does not fully alleviate my concerns in that regard.

**Limitations:**

Yes

**Quality:**

3

**Strengths And Weaknesses:**

### Strengths

- The paper introduces a channel-sensitivity metric, a concatenation-based layer merge, and a hierarchical distillation scheme, providing a well-structured solution to common limitations in LLM pruning.

- CoMe achieves top performance across seven benchmarks and multiple model scales.

- The authors test CoMe's sensitivity to calibration data size, merge granularity, and data distribution mismatch. The method shows stable performance and favorable perplexity compared to baselines.

- CoMe evaluates pruning at the channel level using a custom sensitivity metric and extends the Block Influence score across layers, improving information retention over prior techniques.

---

### Weaknesses

- The evaluation is limited to multiple-choice benchmarks such as ARC, PIQA, and MMLU. While these benchmarks are commonly used to assess factual knowledge and reasoning, they do not capture important aspects of language modeling such as open-ended generation, contextual coherence, or task generalization. As a result, it is difficult to determine whether the proposed compression method preserves the full capabilities of the original model. Including a more diverse set of benchmarks that reflect real-world downstream tasks would provide a more comprehensive evaluation.

- While the proposed channel sensitivity metric offers an intuitive formulation based on expected output perturbation when individual channels are ablated, it remains fundamentally heuristic and lacks rigorous theoretical grounding. The paper does not explain well why this specific formulation is preferable over well-established alternatives such as gradient-based (for example, Taylor or saliency) or second-order importance metrics.

- Ablation studies in the paper show that the heuristic fusion ratio is not consistently effective. As reported in the authors’ own analysis (page 8), the optimal ratio varies across different layer structures, and fixed heuristics may lead to performance degradation.

---

> ### Author Rebuttal · Authors · 2025-07-30
>
> Thank you for taking the time to review our manuscript. We greatly appreciate your valuable feedback. Below, we provide our responses to your comments:
> ## **[Weakness 1 \& Question 1] Benchmark Diversity and Evaluation Scope in Layer Pruning**
> >Thank you very much for your thoughtful suggestions. We agree that evaluating on open-ended generation and task generalization benchmarks would provide a more comprehensive assessment of model performance. However, in the domain of layer pruning, it is standard practice to use pretrained models as the base for pruning. As the number of layers is reduced, the model’s feature representations are often quickly degraded, making fine-grained evaluations such as open-ended generation, contextual coherence, or generalization tests less meaningful. The resulting scores tend to approach random chance. Therefore, in the layer pruning literature, two main evaluation schemes are commonly adopted. The first is measuring PPL on datasets *[1,2]* such as WikiText2 and C4, which reflects the coarse-grained language modeling capability of the pruned model. The second is to use a range of multiple-choice benchmarks (e.g., SuperGLUE, ARC, MMLU) *[3]*, which serve to assess the model’s natural language understanding ability after pruning. For a fair comparison with prior work, we report both types of evaluation results in our paper.
>
> ## **[Weakness 2 \& Question 2] Rationale for Channel Sensitivity Metric**
> >Our choice of the channel sensitivity metric is motivated by prior work and empirical analysis. Specifically, we refer to Wanda *[4]* and MOE-PRUNER *[5]*, which focus on weight-level sparsity and the importance of individual weights. In contrast, CoMe assesses channel importance by considering both weight magnitude and the influence of input activations on layer output, inspired by findings that large activations tend to have a greater impact on model output *[6]*. We provide relevant explanations in Section 4.1 of our manuscript.
>
> >Moreover, we chose not to adopt gradient-based (first-order or second-order) importance metrics for two main reasons: (1) Methods such as Magnitude and Taylor, which rely on gradients, often result in the removal of initial or final layers. To ensure fair comparison, we prohibit pruning of the first and last several layers for these methods (see Appendix A), highlighting the suboptimality of gradient-based approaches for layer pruning. (2) Gradient-based methods require additional storage for gradient information during pruning, which increases computational and memory overhead, contrary to our goal of lightweight pruning and training under resource constraints.
>
> ## **[Regarding Weakness 3 \& Question 3] Selection and Impact of the Channel Retention Ratio**
> >We propose an adaptive parameter fusion ratio formula (Equation 5), which automatically adjusts the fusion ratio between adjacent layers based on their BI scores. Layers with higher BI scores are considered more important and thus retain more parameters. Within the same layer, the FFN and MHA structures share the same fusion ratio. However, our ablation studies in Figures 10 and 14 reveal that the optimal fusion ratios differ between structures (FFN and MHA). We have addressed this issue in the limitations section (line 357) and provided further analysis on page 8.
>
> >The hyperparameter $p$ is used to control the scaling relationship for parameter retention between more important and less important adjacent layers. A larger $p$ means that more important layers retain a higher proportion of parameters. We conduct an ablation study on $p$ in Figure 7 and provide analysis in lines 312–319. Our results show that as $p$ approaches infinity, the pruned model’s performance degrades and converges to that of the DLP method. In other words, an inappropriate choice of $p$ can lead to performance as poor as DLP. Since the optimal value of $p$ varies across different models, we use hyperparameter search to determine the most suitable $p$ for each model. Currently, we do not have a universal setting for $p$ across different models, which is why we have highlighted this limitation in our manuscript.
>
> >In future work, we plan to investigate more effective and adaptive fusion ratios tailored to different structures, layers, and models.
>
> References:
> >*[1] Song, Jiwon, et al. "SLEB: Streamlining LLMs through Redundancy Verification and Elimination of Transformer Blocks." Forty-first International Conference on Machine Learning.*
>
> >*[2] Pei, Zehua, et al. "FuseGPT: Learnable Layers Fusion of Generative Pre-trained Transformers." arXiv preprint arXiv:2411.14507 (2024).*
>
> >*[3] Men, Xin, et al. "ShortGPT: Layers in Large Language Models are More Redundant Than You Expect." Findings of the Association for Computational Linguistics: ACL 2025, edited by Wanxiang Che et al., Association for Computational Linguistics, 2025, pp. 20192–20204.*
>
> >*[4] Sun, Mingjie, et al. "A Simple and Effective Pruning Approach for Large Language Models." The Twelfth International Conference on Learning Representations, 2024.*
>
> >*[5] Xie, Yanyue, et al. "Moe-pruner: Pruning mixture-of-experts large language model using the hints from its router." arXiv preprint arXiv:2410.12013 (2024).*
>
> >*[6] Sun, Mingjie, et al. "Massive Activations in Large Language Models." First Conference on Language Modeling, 2024.*
>
> We hope these responses can address your concerns. If you have further questions, we would be happy to provide additional clarification.

---

> > ### Comment · Reviewer_8TsA · 2025-07-31
> >
> > The author's explanation effectively addresses my concern regarding the diversity of benchmarks and the evaluation scope in Layer Pruning. I find this clarification convincing and therefore have decided to raise my score.
> >
> > However, I believe that Weakness 3 still requires additional experimental support. The current response does not fully alleviate my concerns in that regard.

---

> ### Author Response · Authors · 2025-08-02
> **Additional Evidence and Adaptive Strategies for Layer Fusion Ratios**
>
> We sincerely appreciate your positive feedback and for raising the score.
>
> To further address your concerns about the effectiveness of the heuristic parameter fusion formula, we drew inspiration from SLEB and LaCo to develop a posterior approach for determining fusion weights across different models and layers.
>
> **Implementation details:**
> >The rest of the CoMe computation process remains unchanged. We replace the heuristic fusion ratio scheme with the posterior fusion ratio. Specifically, when merging two layers, we examine fusion ratios from $0:1,\: 0.05:0.95,\: ...,\: 1:0$ and compute the cumulative loss of the pruned model on the calibration dataset for each ratio. We then select the fusion ratio that results in the lowest cumulative loss to fuse layers.
>
> **Analysis:**
> >As shown in the table below, “Heuristic” refers to the original heuristic fusion ratio calculation method in CoMe, while “Posterior” indicates our newly proposed posterior approach.
> Notably, the “Posterior” method does not require any additional hyperparameter tuning for layer fusion. Instead, it adaptively chooses the optimal fusion ratio between layers for different models. During the fusion of two layers, the “Posterior” method evaluates the loss on the calibration dataset 21 times, which is more computationally efficient than SLEB (where the number of loss evaluations of SLEB depends on the remaining layers, typically more than 22. For example, after pruning 30%, LLaMA2-7b retains 22 layers, and LLaMA2-13b retains 28 layers).
>
> >The “Posterior” method further improves CoMe’s performance, reducing the PPL of pruned models on the Wiki-2 and C4, and achieves performance comparable to the “Heuristic” method on benchmarks. The result supports our observation that the optimal fusion ratios vary across different layers. Meanwhile, the “Heuristic” method, which links the fusion ratio to the BI score, incurs only a limited performance loss. Both the “Posterior” and “Heuristic” methods outperform existing SOTA and demonstrate greater stability.
>
> It is worth noting that there are multiple approaches to calculating layer fusion ratios, which do not affect the novelty of our work. Improved strategies could further enhance CoMe’s performance. More importantly, the main innovation of CoMe lies in its substructure concatenation-based fusion method.
>
> Thank you again for your valuable comments and suggestions. We hope this additional explanation further addresses your concerns, and we are happy to discuss any further questions you may have.
>
> |||ARC-c|ARC-e|HellaS|OBQA|PIQA|WinoG|MMLU|Avg|Wiki-2|C4|
> |-|-|-|-|-|-|-|-|-|-|-|-|
> |LLaMA2-7b|ShortGPT|31.91|47.39|45.96|34.80|63.28|61.48|38.66|46.21|49.56|54.92|
> ||SLEB|30.80|51.81|54.07|32.80|68.44|54.14|25.10|45.31|**13.84**|**17.43**|
> ||MKA|34.04|49.58|48.12|35.00|63.00|59.12|**35.64**|46.36|455.34|810.04|
> ||LaCo|30.97|49.79|50.14|35.00|68.34|53.91|24.84|44.71|42.67|39.18|
> ||CoMe-**Heuristic**|**35.24**|54.46|56.56|**35.40**|**68.88**|**61.17**|25.50|48.17|16.53|19.93|
> ||CoMe-**Posterior**|34.64|**54.55**|**58.35**|**35.40**|67.95|61.09|26.89|**48.41**|14.74|18.87|
> |||||||||||||
> |LLaMA2-13b|ShortGPT|35.75|52.82|57.94|38.20|69.91|69.06|47.78|53.07|39.61|29.37|
> ||SLEB|34.04|58.59|63.38|38.60|**75.35**|62.35|26.75|51.29|8.69|**11.64**|
> ||MKA|36.95|53.07|48.67|36.00|65.56|60.46|50.72|50.20|632.28|759.76|
> ||LaCo|33.19|51.43|54.88|39.00|68.39|60.77|24.55|47.46|23.81|27.43|
> ||CoMe-**Heuristic**|42.49|**67.05**|**69.87**|**42.60**|73.34|68.98|51.17|**59.36**|8.85|12.64|
> ||CoMe-**Posterior**|**43.43**|66.96|69.38|40.20|73.50|**69.77**|**51.81**|59.29|**8.56**|12.32|
> |||||||||||||
> |Qwen3-4b|ShortGPT|**32.17**|39.81|**45.73**|31.40|62.46|53.28|27.75|41.80|513.82|417.05|
> ||MKA|32.08|38.47|40.95|30.60|61.53|**60.14**|23.43|41.03|3208.73|2123.39|
> ||**Heuristic**|28.67|47.10|43.95|29.60|63.00|51.46|**32.67**|42.35|37.14|56.13|
> ||**Posterior**|27.56|**47.73**|43.41|**32.00**|**63.98**|54.06|29.35|**42.58**|**28.18**|**40.83**|
> |||||||||||||
> |Vicuna-7b|ShortGPT|33.11|48.40|48.95|34.60|64.25|62.90|41.04|47.61|59.84|61.91|
> ||SLEB|32.00|56.10|53.01|33.00|**70.24**|56.20|24.38|46.42|NaN|NaN|
> ||MKA|34.04|48.15|47.02|35.80|61.97|61.17|**47.99**|48.02|660.93|986.16|
> ||CoMe-**Heuristic**|**36.35**|**58.84**|**56.48**|**42.40**|68.66|**62.98**|25.33|**50.15**|18.69|29.55|
> ||CoMe-**Posterior**|34.56|57.37|56.42|36.80|**69.53**|62.51|28.46|49.38|**15.97**|**22.32**|
> |||||||||||||
> |Mistral-7B|Mag+|25.26|32.07|30.74|28.20|54.08|50.28|25.03|35.09|288.66|253.79|
> ||ShortGPT|32.00|29.71|33.56|31.60|57.51|56.91|22.72|37.72|881.51|760.27|
> ||SLEB|30.80|47.69|57.00|**34.20**|68.44|57.22|25.10|45.78|16.46|21.19|
> ||MKA|32.34|35.90|32.46|30.80|54.95|54.70|25.70|38.12|37735.00|33065.64|
> ||**Heuristic**|31.48|51.85|55.81|31.00|68.77|58.56|27.47|46.42|18.32|22.93|
> ||**Posterior**|**33.45**|**59.05**|**58.32**|31.00|**70.35**|**59.69**|**28.14**|**48.57**|**14.53**|**19.19**|
>
> For details, please see Figure 6 and Tables 8,9,10 (space is limited).

---

> > ### Comment · Reviewer_8TsA · 2025-08-04
> >
> > I think this experiment solid and thanks for your work. I decide to keep my current score.

---

> > > ### Author Response · Authors · 2025-08-04
> > > **Response to Reviewer’s Recognition and Further Inquiries**
> > >
> > > We sincerely appreciate your positive evaluation and recognition of our work. Should you have any remaining questions or concerns regarding our manuscript, research, or the innovations presented therein, we would be delighted to engage in further discussion or provide additional clarification.

---

### Official Review · Reviewer_ACeT · 2025-07-02

**Clarity:** 3
**Significance:** 4
**Originality:** 3
**Rating:** 5
**Confidence:** 4

**Summary:**

This paper introduces CoMe, a novel structured compression framework for Large Language Models (LLMs) that addresses key limitations in existing pruning and knowledge distillation methods. The authors propose a progressive layer pruning framework with a Concatenation-based Merging technology and a hierarchical distillation post-training process to reduce model size while minimizing performance degradation

**Questions:**

- Could an adaptive or learned ratio improve results further, and has this been explored?
- Have the authors considered the computational cost implications in real-world deployment scenarios, particularly concerning hierarchical distillation?
- Can the concatenation-based merging approach be generalized or adapted to non-transformer architectures or other modalities beyond NLP?

**Ethical Concerns:**

["NO or VERY MINOR ethics concerns only"]

**Final Justification:**

The authors have addressed my concerns.

**Limitations:**

See strengths and weaknesses

**Paper Formatting Concerns:**

- line 22: should be state of the art
- line 56: the repeated twice.

**Quality:**

3

**Strengths And Weaknesses:**

**Strength**
- The paper empirically demonstrates its superiority in preserving model capacity and hierarchical knowledge.
- CoMe integrates pruning and distillation into a unified, progressive framework, which is crucial given that previous approaches often handle these stages separately, leading to suboptimal knowledge transfer.
- The "Rethinking the Layer-based Structured Pruning" section (Section 3) provides strong motivation for their proposed approach.
- The introduction of a channel-level sensitivity metric provides a more principled basis for fine-grained pruning.

**Weaknesses**
- The channel sensitivity metric and BI/SBI scores rely on a calibration dataset. The impact of the size and representativeness of this dataset on CoMe's performance is briefly discussed (Fig. 8 ), but further analysis of its sensitivity to dataset characteristics could be beneficial (eg, impact of adding noise, reasoning capabilities, etc).
- While the concatenation-based strategy is effective, its reliance on heuristic parameter preservation ratios limits adaptability across diverse model architectures.
- The computational overhead introduced by hierarchical distillation, especially CoMe-sp needs to be discussed,
- Limited exploration on the interpretability of the retained parameters and their roles post-concatenation.

---

> ### Author Rebuttal · Authors · 2025-07-30
>
> Thank you for taking the time to review our manuscript. We greatly appreciate your valuable feedback. Below, we provide our responses to your comments:
>
> ## **[Weakness 1] Impact of Calibrating Dataset Types, Scales, and Adding Noise on CoMe**
> >Thanks very much for your insightful suggestion. Both the BI/SBI metrics and the channel sensitivity metric indeed rely on a calibration dataset. We present the impact of the calibration dataset size and type on CoMe’s performance in Figure 8 and Figure 11, and provide analysis in Lines 324–340. As shown in Figure 11, CoMe is not highly sensitive to the calibration dataset. Different calibration datasets lead to only minor fluctuations in performance, and the average accuracy is largely unaffected in terms of reasoning ability.
> However, in terms of PPL, using pre-trained data (e.g., Wikitext-2, C4, PG19) as the calibration set yields better results than using instruction-tuning datasets in QA format (e.g., MMLU, Alpaca).
> As shown in Figure 8, when the sample size of the Wikitext-2 calibration dataset exceeds 128, the pruning performance of CoMe becomes stable.
>
> >To further investigate the impact of dataset noise on CoMe’s performance, we conducted an additional experiment in which we randomly shuffled the word order in a proportion of sentences in the calibration set, thereby disrupting the linguistic structure. Using the default experimental settings from the main paper (256 calibration samples), we found that shuffling up to 60\% of the sentences does not significantly affect CoMe’s performance. However, when more than 80\% of the sentences are shuffled, the average accuracy drops by more than two points.
>
> | noise ratio | ARC-c | ARC-e | HellaS | OBQA  | PIQA  | WinoG | MMLU  |  Avg  | Wiki-2 |  C4   |
> |:-----------:|:-----:|:-----:|:------:|:-----:|:-----:|:-----:|:-----:|:-----:|:-----:|:-----:|
> |     0.0     | 35.24 | 54.46 | 56.56  | 35.40 | 68.88 | 61.17 | 25.50 | 48.17 | 16.53 | 19.93 |
> |     0.1     | 34.98 | 54.88 | 56.29  | 34.40 | 69.37 | 60.22 | 25.65 | 47.97 | 16.30 | 20.02 |
> |     0.2     | 34.13 | 56.19 | 55.75  | 36.20 | 68.06 | 59.98 | 23.50 | 47.69 | 15.46 | 19.58 |
> |     0.4     | 34.56 | 55.35 | 54.40  | 38.00 | 68.06 | 59.19 | 26.64 | 48.03 | 15.53 | 20.57 |
> |     0.6     | 35.07 | 55.60 | 55.95  | 36.20 | 69.59 | 60.06 | 22.94 | 47.92 | 14.94 | 22.92 |
> |     0.8     | 32.00 | 52.48 | 53.68  | 34.80 | 65.89 | 60.54 | 22.92 | 46.04 | 19.14 | 26.01 |
> |     1.0     | 33.02 | 50.42 | 51.81  | 32.40 | 66.92 | 59.67 | 23.06 | 45.33 | 19.91 | 26.01 |
>
> ## **[Weakness 2 \& Question 1] Limitation of the Heuristic Parameter Preservation Ratios**
> >Thank you for your insightful comments. We fully agree that the use of heuristic parameter preservation ratios poses limitations in terms of adaptability across different structures, as also noted in our Limitations section (Lines 357–359). Our experiments in Figures 10 and 14 demonstrate that the optimal merging ratio differs between MHA and FFN, indicating that a fixed ratio may not be universally optimal. Furthermore, as shown in Figure 7, we investigated the impact of the fusion ratio hyperparameter and found that heavily prioritizing one layer while neglecting others leads to significant performance degradation. These results suggest that adopting adaptive or learned fusion ratios for different structures could potentially yield better results than our current heuristic approach. We consider this an important direction for future work and plan to explore adaptive strategies in subsequent research.
>
> ## **[Weakness 3 \& Question 2] Analysis of Computational Overhead Introduced by Hierarchical Distillation**
> >Our post-training hierarchical distillation strategy is based on the layer pruning information obtained from the first stage of CoMe. As a result, the number of optimized layers in CoMe-mp is not fixed. In CoMe-mp, each sub-step only updates the parameters for a single layer, whereas in CoMe-sp, layers that are not involved in the merging process during layer pruning (i.e., layers between merged layers) are kept frozen but still included in the training computation. This makes it challenging to directly compare the computational overhead of CoMe-sp and CoMe-mp. In Appendix D, we analyze the differences between CoMe-mp and CoMe-sp and provide a rough comparison of their computational costs. In Table 7, we further compare the number of tokens used by CoMe and single-layer updating methods. Specifically, the computational cost of CoMe-mp is 74\% that of LLM-Streamline and 50\% that of FuseGPT. Under our default experimental settings, for LLaMa2-7B pruned with CoMe, CoMe-sp incurs over 7 times the computational cost of CoMe-mp in terms of parameter updates and forward passes, but requires fewer training tokens (about 70\% of CoMe-mp). Overall, CoMe-sp requires more than 5 times the computational cost of CoMe-mp, but achieves better performance (see Table 1).
> Compared to full fine-tuning, the computational cost of CoMe-mp is approximately $1/M$, where $M$ is the number of layers in the model.
>
> ## **[Weakness 4] Theoretical Basis of Concatenation-Based Merging**
> >The theoretical foundation for parameter retention in CoMe is based on two main aspects, as discussed in Section 3 of our manuscript.
>
> >**First**, the DLP method demonstrates that pruning any single adjacent layer results in minimal performance degradation, which is elaborated in Line 127 (Core Issue 1). This observation led us to consider that while some important feature-processing modules may be discarded during pruning, the model’s intermediate representations can still be effectively processed by subsequent layers even after one layer is removed.
>
> >**Second**, the WSLP method, which blindly merges the weights of adjacent layers, tends to result in over-smoothed parameter distributions. This issue is discussed in Line 142 (Core Issue 2). To address this, we decompose each layer into finer-grained submodules, such as channels in the FFN structure and heads in the MHA structure. These submodules are characterized by the property that adding or removing some of them does not affect the input and output of the remaining modules. Building on this insight, we split layers into feature-mapping submodules and concatenate the submodules from adjacent layers. In this way, the input and output domain distributions of each submodule remain unchanged, and the more important submodules are preserved. This forms the basis of the “Layer as Puzzle Pieces” concept in our title, highlighting that, compared to existing layer pruning methods (DLP and WSLP), CoMe provides a novel framework for layer merging. Our approach addresses layer pruning at a finer granularity and yields a pruned model structure that is more hardware-friendly and easier to deploy.
>
> >We present ablation studies comparing WSLP and various CoMe merging schemes in Table 2, with detailed analyses provided in Section 5.4 and Appendix E. These results demonstrate that our method effectively resolves the issue of mismatched weight mapping between different layers in WSLP. Furthermore, the performance improvements over DLP indicate that CoMe is more effective than both WSLP and DLP.
>
> ## **[Question 3] Generalization of CoMe**
> >In principle, CoMe can be applied to other model architectures. As discussed in our response to [Weakness 4], the CoMe approach decomposes each layer into submodules whose inputs and outputs are independent of one another, and then merges these submodules. Therefore, as long as a model architecture possesses such modularity, CoMe can be adapted for model merging, layer merging, or the fusion of other decomposable structures.
>
> >For example, with minor modifications, CoMe can be extended to convolutional neural networks (CNNs) by pruning layers through the concatenation of convolutional kernels from adjacent layers.
>
> We hope these responses can address your concerns. If you have further questions, we would be happy to provide additional clarification.

---

> > ### Comment · Reviewer_ACeT · 2025-08-05
> > **Response to reviews**
> >
> > I thank the authors for their detailed response. I suggest the authors add a discussion about the overhead of the proposed method, as it can be a disadvantage in different scenarios. I think all my concerns have been addressed.

---

> > > ### Author Response · Authors · 2025-08-06
> > > **Comparison of Post-Training  Overhead for Pruning Methods**
> > >
> > > We sincerely appreciate the valuable suggestions. We will include a more detailed discussion of the post-training computational costs in the final version.
> > >
> > > ## Comparison of Post-Training Computational Costs for Pruning Methods
> > >
> > > >**The computational cost of post-training strategies specifically designed for pruning methods often closely relates to the pruning approach itself, and due to variations in implementation, it is challenging to make a completely fair comparison. Below, we present a comparison of the computational costs of four post-training methods under a fixed setting.**
> > >
> > > >CoMe, LLM-Streamline, and FuseGPT require forward passes to compute hidden features for layer-wise fine-tuning. The cost depends on which layers are pruned, as fine-tuning deeper layers requires more resources to obtain their inputs and outputs. For this analysis, we assume all four methods incur similar costs for this step and **focus solely on the computational cost associated with weight fine-tuning**.
> > >
> > > >**CoMe-mp, FuseGPT, and LLM-Streamline fine-tune one layer per iteration. Therefore, we use the cumulative number of tokens processed during fine-tuning as a proxy for computational cost**. CoMe-sp fine-tunes multiple layers simultaneously, and since not all fine-tuned layers are necessarily adjacent, some non-fused layers may also participate in the forward pass, resulting in slightly higher computational costs than those reported below.
> > >
> > > >We report the total number of tokens processed during the post-training phase when pruning 10 layers from the LLaMA2-7b model. ‘# Steps’ denotes the number of steps per epoch, and ‘# Layers’ indicates the number of layers updated per iteration.
> > >
> > > * *LLM-Streamline*: Fine-tunes one layer per iteration for 5 epochs with 938 steps per epoch. Thus, the cumulative number of tokens is:
> > >
> > > $1 \text{(\\# Iterations)} \times 1 \text{(\\# Layers)} \times 5 \text{(\\# Epoch)} \times 938 \text{(\\# Steps)} \times 32 \text{(\\# Batch size)} \times 2048 \text{(\\# Sequence Length)} \approx 0.31 \text{B} $
> > >
> > > * *FuseGPT*: When pruning 10 layers, performs 10 fusion iterations, updating 7 layers in an iteration, across 20 epochs with 128 steps per epoch. The cumulative number of tokens is:
> > >
> > > $10 \text{(\\# Iterations)} \times 7 \text{(\\# Layers)} \times 20 \text{(\\# Epoch)} \times 128 \text{(\\# Steps)} \times 8 \text{(\\# Batch size)} \times 2048 \text{(\\# Sequence Length)} \approx 2.93 \text{B} $
> > >
> > > * *CoMe-mp*: After pruning, 7 layers are updated, with one layer per iteration, over one epoch with 2000 steps per epoch:
> > >
> > > $7 \text{(\\# Iterations)} \times 1 \text{(\\# Layers)} \times 1 \text{(\\# Epoch)} \times 2000 \text{(\\# Steps)} \times 32 \text{(\\# Batch size)} \times 512 \text{(\\# Sequence Length)} \approx 0.23 \text{B} $
> > >
> > > * *CoMe-sp*: Fine-tunes 7 layers in a single iteration, over one epoch with 10,000 steps:
> > >
> > > $1 \text{(\\# Iterations)} \times 7 \text{(\\# Layers)} \times 1 \text{(\\# Epoch)} \times 10,000 \text{(\\# Steps)} \times 32 \text{(\\# Batch size)} \times 512 \text{(\\# Sequence Length)} \approx 1.15 \text{B} $
> > >
> > > >In practice, the number of layers requiring updates after pruning 10 layers with CoMe may range from 1 to 10, leading to different levels of resource consumption.
> > >
> > > >From these calculations, the order of computational cost is: FuseGPT > CoMe-sp > LLM-Streamline > CoMe-mp. As shown in Table 1, the ranking of model performance is: CoMe-mp > CoMe-sp > LLM-Streamline > FuseGPT. **CoMe-mp achieves the best benchmark performance with the lowest post-training cost, outperforming both FuseGPT and LLM-Streamline. Although CoMe-sp requires more computational resources, its post-training performance is significantly better than that of the other three methods**.

---

### Official Review · Reviewer_SDDe · 2025-07-02

**Clarity:** 2
**Significance:** 3
**Originality:** 2
**Rating:** 4
**Confidence:** 3

**Summary:**

This paper introduces CoMe, a new framework for structured compression of Large Language Models that addresses the limitations of traditional layer pruning and linear weight merging. Instead of removing layers directly, CoMe employs a channel sensitivity metric to identify important channels based on activation intensity and weight norms. These channels are then merged via concatenation across adjacent layers, preserving useful information and enabling progressive pruning.

**Questions:**

1.	All figure used raster image instead of vector image.
2.	I’m confused about the concept of “adjacent layers”. For example, if W^l refer to one up_proj, do W^(l+1) refer to next transformer block’s up_proj or this block’s down_proj?
3.	Why does it called “Hierarchical” Distillation Strategy? In Table 1, I found “sp” performs better. And Equation 8 includes all KL divergence in one loss. I didn’t find any hierarchical distillation.
4.	m in Figure 1 is just 1 in practical right? Because only two layer are merged in one time?

**Ethical Concerns:**

["NO or VERY MINOR ethics concerns only"]

**Final Justification:**

Thanks for the authors' response. Most of my concerns have been addressed. I would like to raise my score.

**Limitations:**

Addressed

**Paper Formatting Concerns:**

NaN

**Quality:**

2

**Strengths And Weaknesses:**

Strengths:

1.	This paper focused on an important problem of structured pruning for model compression.
2.	This paper propose a framework to iteratively merge layers.

Weakness:

1.	The concatenation-based weight merge process in Figure 5 doesn’t seem make sense to me. The concatenation disrupts the channel index of original weight and new weight. Then the new input completely mismatches the original input. I think there lacks some theoretical explanation or motivation about the effectiveness of this new algorithm and why this should be better than others.
2.	Please check “Question”.

---

> ### Author Rebuttal · Authors · 2025-07-30
>
> Thank you for taking the time to review our manuscript. We greatly appreciate your valuable feedback.
>
> Below, we first address [Question 2] and [Question 4] to set the stage for reiterating our method and resolving the concerns you mentioned under weaknesses. We then explain the meaning of the "Hierarchical" Distillation Strategy and address the remaining minor issues.
>
> ## **[Question 2] Clarification on "Adjacent Layers"**
> >In our manuscript, the term “adjacent layers” refers to consecutive layers within the model that are directly connected through input-output relationships. This is a widely used term in academic literature. For example, if the output of layer $l$ serves as the input to layer $l+1$, and the output of layer $l+1$ serves as the input to layer $l+2$, then layers $l$ and $l+1$ are considered “adjacent layers.” Similarly, layers $l$, $l+1$, and $l+2$ (or more consecutive layers) are referred to as “a group of adjacent layers.”
> In Figure 5, $W^{(l)}$ and $W^{(l+1)}$ denote the weight matrices with the same function name (e.g., up\_proj) in adjacent layers.
> Specifically, $W^{(l)}$ corresponds to the up\_proj weight in layer $l$, while $W^{(l+1)}$ corresponds to the up\_proj weight in layer $l+1$.
> Figure 5 illustrates only the fusion method. Depending on the type of matrices, the fusion needs to be performed along different channels (either rows or columns).
>
> >To reduce potential misunderstandings, we will revise the caption of Figure 5 in future versions to:
> “Concatenation-based weight merge process between weight matrices with the same function name in adjacent layers. $W^{(l)}$ and $W^{(l+1)}$ denote the weight matrices with the same function name in two consecutive layers.”
>
> ## **[Question 4] Explanation of Hyperparameter $m$**
> >The hyperparameter $m$ in Figure 1 can be any value less than or equal to the total number of layers in the model. CoMe is capable of merging an arbitrary number ($m$) of layers simultaneously.
> In Figure 9, we conduct an ablation study on $m$, and the results show that merging too many layers at once degrades the performance of CoMe. Therefore, as stated in the experimental details (line 582), we default to merging two layers at a time.
>
> >To reduce potential misunderstandings, we will revise the manuscript as follows:
> At the first mention of $m$ (line 207), we will add the clarification "$m>=2$ and less than or equal to the total number of layers in the model.” In the implementation details (line 269), we will specify "During the layer pruning process, the merging layers number $m$ in each iteration is set to 2.”
>
> ## **[Weakness 1]**
> > We thank the reviewer for pointing out this issue. In future versions of the manuscript, we will include a more detailed theoretical analysis in the Appendix to further clarify our method.
>
> ### * **Explanation of Channel alignment**
> >When designing CoMe, we carefully considered the challenge of channel alignment in the layer. Below, we clarify the intrinsic rationale of the CoMe and explain why it does not lead to a mismatch between the original and the new input.
> We decompose the transformer into two main components (Appendix C): the FFN and MHA. In the FFN, we define the input as $X \in \mathbb{R}^{d}$, the intermediate feature as $I \in \mathbb{R}^{r}$, and the output as $O \in \mathbb{R}^{d}$. The weight matrices $W^{U} \in \mathbb{R}^{d \times r}$, $W^{G} \in \mathbb{R}^{d \times r}$, and $W^{D} \in \mathbb{R}^{r \times d}$ correspond to up\_proj, gate\_proj, and down\_proj, respectively. Thus, the FFN can be formulated as:
>
> >$$ I_{[i]} = \sigma(W^{G}\_{[i,:]} X) (W^{U}\_{[i,:]} X)$$
> >$$ O = \sum W^{D}\_{[:,i]} I\_{[i]} + X$$
>
> >From the above equations, it is evident that permuting the channel indices ($i$) of the intermediate feature $I$ does not affect the channel indices of the input $X$ or the output $O$. It only changes the internal channel order within the FFN.
> When we concatenate the intermediate features from different layers at the corresponding channel indices, we are simply reorganizing the submodules responsible for feature processing within the FFN.
> Regarding channel mapping in the model, we strictly follow the detailed analysis of channel alignment provided in LLM-Pruner *[1]* (Line 612). Therefore, we partition and concatenate the channels in the FFN according to the intermediate size dimension.
>
> >For the same reason, we believe that the WSLP, which directly fuses weights, may result in incorrect weight fusion, as it does not ensure channel correspondence across different layers, potentially leading to smoothed weight distributions.
>
> >Similarly, for the MHA structure, we partition channels based on the input dimension of O\_proj. Since the head structure in MHA is inherently indivisible, we further treat each head (comprising $W^Q$, $W^K$, and $W^V$) as a “channel group.” Changing the order of heads or the input order of O\_proj does not affect the overall input or output of the MHA.
>
>
> ### * **Theoretical Basis of Concatenation-Based Merging**
> >Layer pruning achieves higher efficiency by reducing the number of layers, which inevitably leads to input space shifts between layers. This issue is common to DLP, WSLP. The key lies in mitigating such misalignments.
>
> >The core idea of CoMe is to divide each layer into finer-grained functional blocks. From the perspective of the FFN module, after partitioning parameters into more granular channels, each channel can be regarded as a small expert. Fusing two adjacent layers is thus equivalent to selecting and retaining the most important experts. Several recent works *[2,3]* have adopted similar perspectives to analyze the FFN (or experts). Likewise, in the MHA structure, each head can naturally be viewed as an independent feature processor.
>
> >Furthermore, we employ a Channel Sensitivity Metric to select which channels to retain or discard based on their contribution to the output. Channels with larger products of activation and weight magnitude are considered to have a greater impact, which avoids the need to compute gradients. This approach is consistent with the perspectives of several model sparsification methods *[4,5]*.
>
> ## **[Question 3] Definition of "Hierarchical" Distillation**
> >We provide a detailed description of the Hierarchical Distillation Strategy in Section 4.3 of our manuscript. Lines 238–245 specifically explain how the CoMe-mp process is transformed into CoMe-sp by modifying the training approach and loss function. **Our hierarchical distillation strategy leverages layer correspondences $\mathcal{P} = \\{\\{a_1, b_1\\}, ..., \\{a_N, b_N\\}\\}$ established during pruning**, where $a_i$ and $b_i$ are indices of corresponding layers in the teacher and student models, respectively. For example, if $\mathcal{P} = \\{\\{1, 3\\}, \\{2, 6\\}, \\{5, 8\\}\\}$, then the output of the 1st layer in the student is aligned with the output of the 3rd layer in the teacher, the 2nd layer in the student is aligned with the 6th layer in the teacher, and the 5th layer in the student is aligned with the 8th layer in the teacher.
> The loss function (Equation 8 in the manuscript):
> >$$\mathcal{L}\_{\text{KL-sp}} = \frac{1}{|\mathcal{P}|} \sum\_{\{a,b\} \in \mathcal{P}} \mathbb{E}\_{\mathcal{D}\_{\text{train}}} \left[ D\_{\text{KL}} \left( \sigma(H^{(t,a)}) \| \sigma(H^{(s,b)}) \right) \right]$$
> enforces alignment of **layer-wise features** across all hierarchical levels, rather than only the final logits.
>
> >**Therefore, both CoMe-mp and CoMe-sp distill knowledge from the original model to restore the performance of the pruned model by aligning the outputs at the “layer" level. For this reason, our approach is referred to as a “Hierarchical Distillation Strategy.”**
>
> ## **[Question 1] Format of Figures**
> >Our manuscript was prepared using Overleaf, with all images provided in .pdf format. Nevertheless, following your helpful comments regarding the quality of our figures, we also noticed that some images appear blurry when zoomed in. We will provide clearer vector images in future versions of the manuscript.
>
> ## Reference
> >*[1] Ma, Xinyin, Gongfan Fang, and Xinchao Wang. "Llm-pruner: On the structural pruning of large language models." Advances in neural information processing systems 36 (2023): 21702-21720.*
>
> >*[2] Lo, Ka Man, et al. "A Closer Look into Mixture-of-Experts in Large Language Models." Findings of the Association for Computational Linguistics: NAACL 2025. 2025.*
>
> >*[3] Zhu, Tong, et al. "LLaMA-MoE: Building Mixture-of-Experts from LLaMA with Continual Pre-Training." Proceedings of the 2024 Conference on Empirical Methods in Natural Language Processing. 2024.*
>
> >*[4] Sun, Mingjie, et al. "A Simple and Effective Pruning Approach for Large Language Models." The Twelfth International Conference on Learning Representations, 2024.*
>
> >*[5] Xie, Yanyue, et al. "Moe-pruner: Pruning mixture-of-experts large language model using the hints from its router." arXiv preprint arXiv:2410.12013 (2024).*
>
> Thank you again for your valuable feedback, and we hope our responses address your concerns. If you have further questions, we would be happy to provide additional clarification.

---

> > ### Comment · Reviewer_SDDe · 2025-08-05
> > **Official comment by Reviewer SDDe**
> >
> > Thanks for the authors' response. Most of my concerns have been addressed. I would like to raise my score.

---

> > > ### Author Response · Authors · 2025-08-05
> > > **Appreciation for the Reviewer’s Recognition**
> > >
> > > We sincerely appreciate your thorough review and positive recognition of our work. Should you have any additional questions or concerns regarding our manuscript, research, or the innovations presented, we would be pleased to engage in further discussion or provide any necessary clarifications.

---

### Author Response · Authors · 2025-08-09
**Summary Rebuttal**

Dear Area Chair,


We sincerely appreciate the valuable time and efforts of the AC and the reviewers. We are pleased that the reviewers recognize the novelty and originality of our framework (Reviewer #SDDe, Reviewer #1RXD), the significance and impact of our contributions (Reviewer #ACeT, Reviewer #8TsA), the methodological rigor and thorough experimental validation (Reviewer #SDDe, Reviewer #8TsA), and the effectiveness and robustness of CoMe across diverse models and benchmarks (all reviewers). Below, we summarize how we have addressed each reviewer’s concerns in detail:

**Reviewer #SDDe:**
>We provided additional theoretical analysis to clarify the rationale behind our method’s design and reiterated certain statements for clarity. The reviewer indicated that most concerns have been addressed and decided to raise their score ($score >= 4$).

**Reviewer #ACeT:**
>We added robustness analysis and computational overhead comparisons, demonstrating the efficiency of CoMe. We also clarified the cross-architecture generalization capability of our approach. The reviewer acknowledged that all concerns were resolved ($score = 4$).

**Reviewer #8TsA:**
>We explained our benchmark selection and its diversity, clarified the rationale and design of the channel sensitivity metric, and highlighted its advantages over gradient-based methods. Furthermore, we proposed a posterior adaptive scheme to address the limitations noted in the manuscript. The reviewer confirmed satisfaction with the additional analyses and decided to raise their score ($score >= 4$).

**Reviewer #1RXD:**
>We clarified the differences between CoMe and LaCo, explained the theoretical underpinnings of CoMe, and further distinguished CoMe from LoRA, including the reasons for their non-comparability. We also provided a rough comparison of computational overhead with other methods. We are currently awaiting this reviewer’s response.

We hope this summary demonstrates our efforts to address all reviewer feedback and highlights the improvements made to our manuscript during the rebuttal process.

Finally, we would like to express our sincere gratitude to you and the reviewers for your constructive feedback, which has contributed to a significantly improved version of our work.

Best regards,
The authors

---

### Decision · Program_Chairs · 2025-09-17

**Decision:**

Accept (poster)

**Comment:**

This paper proposes CoMe, a layer pruning framework for large language models that addresses limitations of direct layer removal and linear weight layer fusion. CoMe selects channels from adjacent layers for weight merging and applies knowledge distillation to restore performance. Most reviewers gave positive ratings to this paper. Reviewer 1RXD noted that the paper does not provide a fair comparison of computational costs and that LoRA was not included as a baseline. Other Reviewers acknowledged that most of their responses have been resolved during the rebuttal period.

After carefully considering all reviews, the AC believes that the weakness identified by Reviewer 1RXD does not undermine the paper’s main contribution. Since this work focuses on layer pruning and merging, it is not necessary to compare with PEFT methods such as LoRA.

The authors are encouraged to incorporate all new results from the rebuttal, along with the reviewers’ suggestions—such as providing a more rigorous comparison of computational costs with other methods—into the camera-ready version.